# A pseudoproxy assessment of why climate field reconstruction methods perform the way they do in time and space

Sooin Yun [1], Jason E. Smerdon [2], Bo Li [1], and Xianyang Zhang [3]

[1]Department of Statistics, University of Illinois at Urbana-Champaign, Champaign, Illinois
[2]Lamont-Doherty Earth Observatory, Columbia University, Palisades, New York
[3]Department of Statistics, Texas A&M University, College Station, Texas

**Correspondence:** Sooin Yun (syun13@illinois.edu)

**Abstract.** Spatiotemporal paleoclimate reconstructions that seek to estimate climate conditions over the last several millennia are derived from multiple climate proxy records (e.g. tree rings, ice cores, corals, and cave formations) that are heterogeneously distributed across land and marine environments. Assessing the skill of the methods used for these reconstructions is critical as a means of understanding the spatiotemporal uncertainties in the derived reconstruction products. Traditional statistical measures

of skill have been applied in past applications, but they often lack formal null hypotheses that incorporate the spatiotemporal characteristics of the fields and allow for formal significance testing. More recent attempts have developed assessment metrics to evaluate the difference of the characteristics between two spatiotemporal fields. We apply these assessment metrics to results from synthetic reconstruction experiments based on multiple climate model simulations to assess the skill of four reconstruction methods. We further interpret the comparisons using analysis of Empirical Orthogonal Functions that represent

the noise-filtered climate field. We demonstrate that the underlying features of a targeted temperature field that can affect the performance of CFRs include: (i) the characteristics of the eigenvalue spectrum, namely the amount of variance captured in the leading EOFs; (ii) the temporal stability of the leading EOFs; (iii) the representation of the climate over the sampling network with respect to the global climate; and (iv) the strength of spatial covariance, i.e. the dominance of teleconnections, in the targeted temperature field. The features of climate models and reconstruction methods identified in this paper demonstrate more

detailed assessments of reconstruction methods and point to important areas of testing and improving real-world reconstruction methods.

## 1 Introduction

    Climate field reconstructions (CFRs) are spatially explicit estimates of past climate conditions that use layered or banded archives containing chemical, biological, or physical indicators as proxies for climate prior to the advent of instrumental

records. CFRs can target climate fields over a range of timescales and mean states, but a particular period of focus for large-scale (continental to global) CFRs has been the Common Era (CE), or the last two millennia (e.g. Jones et al., 2009; Christiansen and Ljungqvist, 2017; Smerdon and Pollack, 2016). This interval contains an abundance of high-resolution proxy records that allow seasonal-to-annual CFRs on regional-to-global spatial scales. Application of CFRs over the CE have provided myriad insights into climate variability and change (see reviews in Jones et al., 2009; Cook et al., 2016; Smerdon, 2017), including,

for example, characterizations of volcanic impacts on climate (Anchukaitis et al., 2017, 2010; Zhu et al., 2020; Wahl et al., 2014; Tejedor et al., 2021a, b), determination of the causal mechanisms of multidecadal droughts in North America (Cook et al., 2004; Coats et al., 2016; Steiger et al., 2019; Cook et al., 2016), characterization of hydroclimatic variability and forced changes on continental scales (e.g. Cook et al., 2004, 2010; Stahle et al., 2016; Palmer et al., 2015; Stahle et al., 2020; Erb et al., 2020), and assessments of model performance (e.g. Smerdon and Coauthors, 2017; Mann et al., 2009b; Coats et al.,
2020).

There are many different CFR methods (e.g. Tingley et al., 2012; Smerdon et al., 2016; Steiger et al., 2014b; Mann et al., 2009b), most of which differ based on the manner in which climatic proxies – principally measurements of a given archival indicator – are transformed to estimate a given climatic quantity and how spatial and temporal covariance estimates are used to infer missing data. CFR methods are in turn applied to a wide range of proxy networks that are often curated for a specific
purpose. These collective and fundamental decisions determine the nature of any given derived CFR, all of which are subject to shared and unique uncertainties tied to their spatial and temporal performance (Wang et al., 2014; Smerdon et al., 2016; Klein et al., 2019). Assessing the spatiotemporal skill of CFR methods is therefore critical as a means of understanding the uncertainties in derived reconstruction products and there has been an ongoing focus in the literature to better understand CFR performance on hemispheric and global scales (Smerdon et al., 2008a, b, 2010a, b, 2011, 2016; Li and Smerdon, 2012;
Dannenberg and Wise, 2013; Steiger et al., 2014a; Evans et al., 2014; Wang et al., 2014; Yun et al., 2020; Harris et al., 2020).

Over the last decade and a half, one approach that has emerged for evaluating CFR methods relies on synthetic exercises called pseudoproxy experiments (PPEs; Smerdon, 2012). The basic premise of PPEs is to subsample a given spatially and temporally complete field from a transient last-millennium simulation derived from a fully-coupled global climate model in a way that mimics the limited instrumental and proxy data available for deriving real-world CFRs. The subsampled data are
then input into a reconstruction algorithm that is used to generate a CFR estimate for a given last-millennium simulation. The derived CFR can then be compared to the withheld and known values of the simulated climate field as a means of evaluating reconstruction skill in both space and time. The advantage of PPEs lies in their ability to establish controlled experimental environments in which the performance of CFR methods can be assessed.

Despite their widespread utility, interpretations of PPEs are complicated by the fact that synthetic pseudoproxies are only an
approximation of the complicated signal and noise structures inherent to proxy records (e.g. Wang et al., 2014; Evans et al., 2013, 2014), while the model-specific climates that underly each PPE may not fully mimic the spatiotemporal characteristics of the real climate system. Hence, Christiansen et al. (2009) emphasized the importance of the underlying spatiotemporal characteristics of the target field by accessing the multi-model dependencies through a means of making surrogate climates and Smerdon et al. (2011) tested the spatiotemporal skill of four CFR methods in the context of a global PPE based on last-
millennium simulations from two climate models. This work was later expanded by Smerdon et al. (2016) to test four CFR methods using newer last-millennium simulations from five models that contributed to the Coupled Model Intercomparison Project Phase 5 and the Paleoclimate Modeling Intercomparison Project Phase 3 (CMIP5/PMIP3) in support of Assessment Report 5 of the Intergovernmental Panel on Climate Change (Stocker et al., 2013). The findings of both studies highlighted important differences between the performance of the employed methods, while also noting that the reconstruction skill was

dependent on the last-millennium simulation that was the basis of the PPE. It is therefore important to perform PPEs based on multiple last-millennium simulations, while working to understand how the impacts of modeled spatiotemporal climate characteristics translate into implications for CFRs performed for the real climate system. Critically, the skill of real-world CFRs is ultimately dependent on the spatiotemporal character of the actual climate system, thus necessitating interpretations of PPEs in terms of the underlying spatiotemporal characteristics of each climate model simulation on which they are based and how these characteristics compare to those of the real climate system.

Improved interpretations of PPEs that take into account the above considerations require improved and more detailed skill assessments. Almost all skill characterizations of previous PPEs are descriptive in nature, largely employing spatial maps and global aggregates of statistics such as the mean biases in derived CFRs, correlations between the CFRs and known fields, or the root mean square error of the CFRs relative to the known fields. While such comparisons are useful for evaluating the relative performance of the various CFR methods, they do not employ a formal null hypothesis that can determine whether or not the spatiotemporal differences between reconstructed fields are statistically significant. One limitation this presents, for example, is an assessment of whether one method in a PPE performs better than another in a statistically robust sense, or whether spatiotemporal differences among methods are simply due to random error. An additional challenge of previous statistical assessments is that they interpret the derived CFRs as complete spatiotemporal representations of the targeted climate field, despite the fact that most CFR methods target reduced-space versions of a field by selecting, for instance, only a few leading patterns from matrix decompositions of the field's covariance matrix. Despite such reductions being the basis of almost all CFR approaches, it is rare that skill assessments decompose reconstruction performance in terms of leading reconstructed and targeted spatiotemporal patterns.

In an attempt to more rigorously compare spatiotemporal characteristics of reconstructed and targeted climate fields in PPEs, Li and Smerdon (2012) formalized a null hypothesis for these comparisons. Their approach was expanded by Li et al. (2016) who applied methods for comparing the mean and covariance structure between two spatiotemporal random fields developed by Zhang and Shao (2015). This method has significant advantages over other parametric tests: (i) it evaluates whether the spatially-varying mean or covariance structures of two climate fields exhibit similar patterns; (ii) the noise will be filtered by the principal component analysis; (iii) it is completely non-paramteric and thus free of model-misspecification risks; (iv) allows dependence between two fields and temporal correlation within each data set; and (v) it is constructed to separate skill within a given EOF basis and thus allows assessments of skill within each leading pattern of spatiotemporal variability at different directions or subspaces.

We use the formalism of Li et al. (2016) and the established PPE framework from Smerdon et al. (2016) to evaluate whether there are statistically robust differences between derived CFRs and targeted climate fields. We demonstrate how CFR skill can be separated into leading modes of variability, which allows us to better interpret the performance of each CFR in terms of the particular spatiotemporal characteristics of the climate model simulations on which each PPE is based. This approach allows us to more clearly articulate the reasons why the applied CFR methods perform differently within model-specific PPEs and across PPEs based on different last-millennium simulations. Our results demonstrate how our methods can be used to improve

interpretations of uncertainties and limitations in state-of-the-art CFR methods and provide improved understanding of how specific characteristics of the real climate system may give rise to enhanced or reduced CFR performance.

## 2 Data and Methods

The adopted experimental setup is specifically chosen to be consistent with previous PPE and methodological assessments of Smerdon et al. (2016) and Li et al. (2016). This consistency allows meaningful comparisons to previous results that were either based on more traditional skill assessment metrics or did not fully diagnose the underlying reasons for skill differences using Li et al. (2016) methods. In the following subsections we describe the last-millennium simulations that are used as the basis of our PPEs and the CFR methods employed.

### 2.1 Pseudoproxy Experimental Setup

The PPEs employ concatenated last-millennium (850-1849 CE) and historical simulations (1850-2005 CE) from modeling centers as configured and implemented in CMIP5/PMIP3. Simulations from the following models are employed: the Beijing Climate Center CSM1.1 model (BCC), the National Center for Atmospheric Research Community Climate System Model version 4 (CCSM), the Goddard Institue for Space Studies E2-R model (GISS), the Institue Pierre-Simon Laplace CM5A-LR model (IPSL) and the Max-Plank Institute ESM-LR model (MPI); abbreviations in parentheses are the convention by which each model and associated PPE framework will be referenced hereinafter. In all cases, annual means from the surface temperature fields of the model are used and all fields are interpolated to uniform $5°$ latitude-longitude grids from which all subsampling is performed (Smerdon et al., 2016). The CMIP5/PMIP3 simulations from these models were chosen in Smerdon et al. (2016) based on the availability of PMIP3 last-millennium simulations at the time. There have since become available additional last-millennium simulations, most notably the last-millennium ensemble from the National Center for Atmospheric Research (NCAR) Community Earth System Model (CESM) and a few last millennium simulations from the PMIP4 archive. These and additional simulations will ultimately be available for PPEs, but in the interest of consistency, and because last-millennium simulations from the PMIP4 archive are not yet fully available, we limit our assessment to those simulations that were used for PPEs in Smerdon et al. (2016) and subsequently in Li et al. (2016).

The basic premise of PPEs is to subsample the pseudoproxy and instrumental data from the simulated climate model in a way that approximates their availability in the real world. Each model field is therefore subsampled to approximate available instrumental temperature grids and proxy locations in a given proxy network. The PPE framework employed herein approximates available grids in the Brohan et al. (2006) surface temperature dataset ($M = 1,732$ grid cells) and the locations of the proxies used in the Mann et al. (2009a) CFR (yielding $p = 283$ proxy locations; see Figure 1). Global proxy networks have been expanded since Mann et al. (2009a) such that the sampling schemes used herein slightly underestimate the densest network in state-of-the-art multiproxy datasets (PAGES2k Consortium, 2017). The spatial sampling biases represented in the Mann et al. (2009a) network are nevertheless similar to the PAGES2k biases, with dense sampling in the Northern Hemisphere extratropics, more sampling over land than over oceans, and sparser sampling in the tropics and Southern Hemisphere. Improvements

in the PAGE2k sampling are nevertheless present over some regions, such as the tropical oceans and Antartica. Despite these differences, the descriptions we have just provided are based on the most recent sampling interval in the Mann et al. (2009a) and PAGES2k networks. The adopted pseudoproxy sampling scheme is a best-case scenario of the Mann et al. (2009a) network and similarly would be representative of the denser sampling intervals in the PAGES2k network. For instance, the increased sampling of the tropical oceans in the PAGES2k network, relative to Mann et al. (2009a), is associated with the coral-derived proxies in those regions, but these records typically only span several centuries and are therefore not part of the sampling in the earlier centuries of the CE. Our emulation of the Mann et al. (2009a) network is therefore still applicable to more recent multiproxy compilations and in general represents a best-case sampling scenario even for the most up-to-date networks given that these networks still lose significant numbers of records back in time.

The application of PPEs also requires that the time series subsampled from last-millennium simulations are perturbed with noise to mimic the imperfect connection between measurements in proxy indicators and the climatic signal for which they are interpreted. The common approach within PPEs is to add randomly generated noise series to the subsampled modeled time series representing proxy data, with noise amplitudes scaled to mimic the signal-to-noise ratios (SNRs) that are characteristic of real-world proxies. In this study, we use the CFRs from Smerdon et al. (2016) that were derived from pseudoproxies perturbed with Gaussian white-noise at an SNR of 0.5, a value deemed to be within to the range of SNRs (0.5-0.25) in real-world proxy networks (Smerdon, 2012; Wang et al., 2014). In addition to $SNR = 0.5$, we also analyze a no-noise experiment ($SNR = \infty$). In all model cases, the same realization of 283 Gaussian white-noise series are used to perturb the pseudoproxy network.

The adopted PPE design is overall a simplification of real-world conditions. Real proxies typically include noise that is multivariate (i.e. tied to climatic conditions in addition to temperature), autocorrelated, and non-stationary in time (e.g. Jacoby and D'Arrigo, 1995; Briffa et al., 1998; Esper et al., 2005; Evans et al., 2002; Anchukaitis et al., 2006; Franke et al., 2013; Evans et al., 2014; Baek et al., 2017; Anchukaitis et al., 2017; Wilson et al., 2016), while most proxies typically respond to season-specific conditions (e.g. Pauling et al., 2003; Anchukaitis et al., 2006; St. George et al., 2010; Baek et al., 2017). Real-world spatiotemporal variability is also assumed to be realistically represented in the modeled climates used as the basis of the PPEs. Important characteristics of the modeled climates nevertheless can vary across simulations and stand in contrast to observed behavior, such as the strength and spatial covariance of teleconnections (e.g. Coats et al., 2013). We therefore interpret our PPE design as a best-case scenario, relative to real-world conditions, and additional modifications to the PPE framework to more fully mimic reality is expected to only further degrade CFR skill (e.g. Von Storch et al., 2004; Von Storch and Stehr, 2006; Mann et al., 2007; Wang et al., 2014; Evans et al., 2014; Smerdon, 2012; Smerdon et al., 2016).

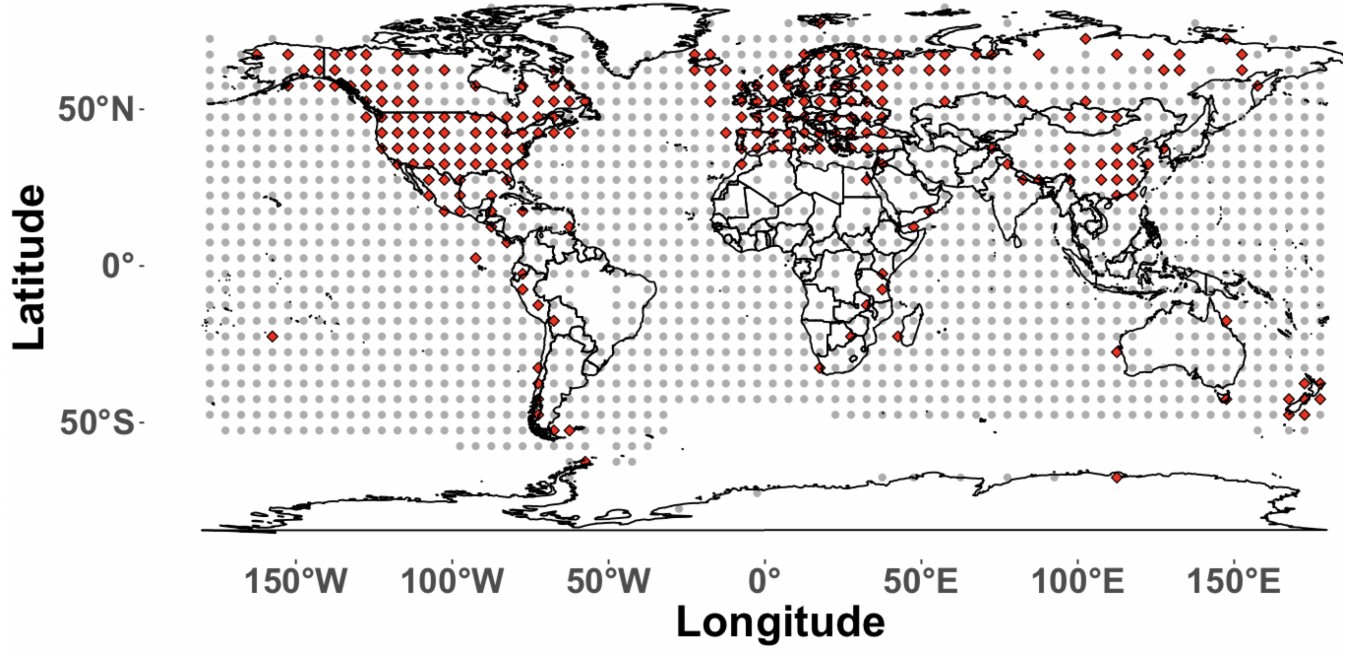

**Figure 1. Proxy Network and Instrumental Sampling Mask.** Grey dots ($M = 1,732$) are the locations where the temperature field is sampled and the red dots indicate the grid points where the temperature locations are sampled to derive the pseudoproxies ($p = 283$).

## 2.2 Climate Field Reconstructions

We analyze four CFR methods that have been widely applied in the CFR literature and specifically discussed in the context of the analyzed PPEs in Smerdon et al. (2016). These methods include two versions of regularized expectation maximization (RegEM) (Schneider, 2001; Mann et al., 2007; Christiansen et al., 2009), standard ridge regressions (Hoerl and Kennard, 1970) and canonical correlation analysis (CCA) (Christiansen et al., 2009; Smerdon et al., 2010b), all of which are briefly described in the following subsections. In general, however, all of the tested methods comprise versions of regularized multivariate

regressions. The basic approach of these methods relates a matrix of climate proxies to a matrix of climate data during a common time interval (termed the calibration interval) using a linear model. As a general example, consider an $m \times n$ matrix, $P$, which contains proxy values, and an $r \times n$ matrix, $T$, consisting of instrumental temperature records where $m$ is the number of proxies, $r$ is the number of spatial locations in the instrumental field, and $n$ corresponds to the number of time steps in the temporal period of overlap between the proxy and instrumental data. The regression of $T$ columns on $P$ columns for

time-standardized matrices ($T_{std}$ and $P_{std}$) with rows that have means of zero and standard deviations of one can be written:

$$T = M_t + S_t T_{std}, \qquad P = M_p + S_p P_{std},$$

where $M_t$ is a matrix of identical columns equal to the average of all columns of the matrix $T$, and $S_t$ is a diagonal matrix with elements that are the standard deviations of the rows of matrix $T$; $M_p$ and $S_p$ are similarly defined for matrix $P$. In these terms,

$$T_{std} = BP_{std} + \varepsilon,$$

where $B$ is a matrix of regression coefficients with dimensions $r \times m$, and $\varepsilon$ is the residual error. According to standard linear regression theory, the error variances of all the elements of $\varepsilon$ are minimized if $B$ is chosen as:

$$B = (T_{std}P_{std}^T)(P_{std}P_{std}^T)^{-1},$$

where the superscript $T$ denotes the matrix transpose. Temperature thus can be predicted, or "reconstructed," using this regression matrix during periods in which proxy data are available:

$$\hat{T} = M_t + S_t B S_p^{-1}(P - M_p),$$

where $\hat{T}$ denotes a matrix of reconstructed temperatures. Most regression-based CFRs differ in the way that $B$ is estimated. As an example, consider a reduced rank representation of $P_{std}$ using singular value decomposition (SVD):

$$P_{std}^r = U_p^r \Sigma_p^r V_p^{r\,T}.$$

Here $P_{std}^r$ denotes the reduced-rank representation of $P_{std}$, and matrices with the superscript $r$ are the truncated versions of the SVD factors. Similarly, the reduced-rank version of $T_{std}$ is written:

$$T_{std}^r = U_t^r \Sigma_t^r V_t^{r\,T}.$$

$P_{std}^r$ and $T_{std}^r$ thus can be substituted into the expression for $B$ as follows,

$$B = U_t^r \Sigma_t^r V_t^{r\,T} V_p^r (\Sigma_p^r)^{-1} U_p^{r\,T}$$

Given this expression for $B$, several choices can be made. The first of which involves rank-reducing either $T$, $P$ or both and by how much, which is the basis of principal component regression. The other choice involves regularization. In the representation of $B$ above, this can by done by reducing or filtering the cross-covariance matrix $V_t^{r\,T} V_p^r$. If we use CCA formalism as an example, the cross-covariance matrix can again be factored using SVD as $O_t^r \Sigma_{cca}^r O_p^{r\,T}$, which is the truncated form of the cross-covariance matrix in which some leading number of canonical coefficients have been retained. Beginning with the original expression for $B$, ridge regression alternatively does not adopt $T$ and $P$ reductions, but instead replaces the inverse matrix in $B$ with $(P_{std}P_{std}^T + h^2 I)^{-1}$, where $h$ is the ridge parameter, as a means of filtering the cross-covariance between $T$ and $P$. Further aspects of these methods are discussed in more detail elsewhere (e.g. Hoerl and Kennard, 1970; Schneider, 2001; Christiansen et al., 2009; Tingley et al., 2012; Smerdon et al., 2010b), but this brief discussion provides the

basic framework for how the regression-based CFR methods are applied. The following subsections provide more details on the specific versions of the methods that we apply and more references to the supporting literature. We also describe directly below the choices for the calibration and reconstruction intervals.

All CFR methods use a calibration from 1850-1995 C.E. and a reconstruction interval from 850-1849 C.E. Temperature and proxy data are available after 1995, but the proxy network as used in Mann et al. (2009a) becomes sparse after 1995 because many records collected over the last several decades obviously do not include measurements after their date of collection. Hence, as in Mann et al. (2009a), Smerdon et al. (2016), and Li et al. (2016), our calibration period is chosen to be 1850-1995 C.E., which follows the convention of previous PPE frameworks. We also note that while we test these specific configurations and methods, the skill assessments that we employ and the methodological insights that are developed are not exclusive to the four methods that are tested because of broad commonalities across the CFR problem.

### 2.2.1   Regularized Expectation Maximization

The RegEM framework is based on regularized multivariate linear regressions, specifically ridge regression (Rutherford et al., 2003, 2005) and truncated total least squares (Mann et al., 2007, 2009b), but the regression coefficients are non-linearly and iteratively estimated by casting CFRs as a missing-value or imputation problem. Within this formalism, the mean and covariance of an incomplete dataset are initially infilled, updated, and ultimately selected based on the minimization of the expected mean-squared error of the infilled data within some specified threshold of convergence. The two versions of RegEM that we use herein both use truncated total least squares for regularization (Schneider, 2001; Mann et al., 2007; Christiansen et al., 2009). The first version is the standard form of RegEM using truncated total least squares as originally described by Schneider (2001), hereinafter TTLS, and the second is the hybrid version applied by Mann et al. (2009a), hereinafter TTLH. The hybrid convention calibrates the multiproxy network on the target temperature field in split spectral domains by first separating the target field and the multiproxy (or pseudoproxy) network into high and low-frequency components. We follow the Mann et al. (2009a) convention by splitting these two domains at the 20-year period using a ten-point butterworth filter. The hybrid reconstruction is then derived by calibrating the pseudoproxy network in the two frequency domains using the RegEM algorithm and subsequently combining the reconstructions from each domain to derive a complete field (see Mann et al., 2005, 2007 for further description of the hybrid method). Note also that differences between reconstructions derived from the hybrid and standard versions of the RegEM method have been reported to be minimal (Rutherford et al., 2005; Mann et al., 2005, 2007; Smerdon et al., 2011), although the importance of hybrid calibrations on the skill of the derived reconstructions has been debated (Rutherford et al., 2010; Christiansen et al., 2010). A linear fit to the log-eigenvalue spectrum is used to determine the truncation parameter for the RegEM CFRs in the same manner that was advocated by Mann et al. (2007) for the high-frequency component of their derived hybrid reconstructions. For the Mann et al. (2009a) CFRs, a linear fit to the log-eigenvalue spectrum was again used to determine the truncation parameter for the high-frequency component of the reconstructions, while the low-frequency truncation was determined by selecting the eigenvalue rank yielding 33% of the cumulative variance in the low-frequency field. This percentage of retained cumulative variance is reduced from 50%, as

originally adopted by Mann et al. (2007); the value of 33% has since been advanced by Rutherford et al. (2010) and Mann et al. (2009a) as more appropriate.

### 2.2.2 Ridge regression

The CFRs derived from ridge regressions used the standard formulation (Hoerl and Kennard, 1970). This approach is a break from earlier studies that have used ridge regression as the form of regularization in the iterative RegEM algorithm, but there are now several studies that have applied the single ridge regression approach in the context of CFRs (Smerdon et al., 2011, 2016). CFRs derived from ridge regressions in these two formulations have been discussed in detail in various publications (Schneider, 2001; Mann et al., 2005; Smerdon and Kaplan, 2007; Lee et al., 2008; Smerdon et al., 2008a, 2010a; Christiansen et al., 2009). The standard ridge regression is applied herein because the iterative RegEM ridge regression converges to the single ridge regression result when missing values comprise a single and regular block in the data matrix, which is true in the context of our PPE design. The selection of the ridge parameter, defined as $h$ in the introduction of subsection 2.2, is determined using the same approach described by Schneider (2001) for ridge-based RegEM, namely by the minimization of the generalized cross validation function (Golub et al., 1979).

### 2.2.3 Canonical correlation analysis

We apply CCA to derive CFRs as described in Smerdon et al. (2010b) and as briefly presented in the introduction of subsection 2.2. SVD is used to factor and truncate the $T$, $P$, and cross-covariance matrix $V_t^{r\,T} V_p^r$. The optimal dimensional reductions for all three of these dimensions was determined using a minimization of the root-mean squared error in a 'leave-half-out' cross-validation scheme, as described by Smerdon et al. (2010b).

## 3 Skill Assessment

### 3.1 A brief review of the functional methods

The methods of comparing two spatiotemporal random fields developed in Zhang and Shao (2015) and Li et al. (2016) are based on a functional data analysis approach. The basic idea is to perform the comparison in subspaces that are of much lower dimension but preserve a large portion of the variability. Moreover, these comparisons can be done on individual EOF-PC pairs, allowing CFR assessments to be done on specific leading modes of the targeted and reconstructed fields. We briefly review the theoretical framework for the skill assessments below.

Let $\{X_t(s)\}_{t=1}^N$ and $\{Y_t(s)\}_{t=1}^N$ be two spatiotemporal random fields observed over spatial locations, $s \in D$, and time points, $t = 1, \ldots, N$. The two random fields can be either independent or dependent. For our CFR assessment, we use $X$ to denote the synthetic climate from climate models and $Y$ the CFRs based on the $X$ process. We define the mean function of each spatial process as: $\mu_X(s) = E\{X_t(s)\}$ and $\mu_Y(s) = E\{Y_t(s)\}$. Also, the covariance function of each spatial process is defined as $C_X(s, s') = cov\{X_t(s), X_t(s')\}$ and $C_Y(s, s') = cov\{Y_t(s), Y_t(s')\}$ over $s$ and $s' \in D$, respectively. We assume second-order

stationarity in time, i.e., the mean at each location is a constant over time, and the spatial covariance function follows the same structure at different times. We do not assume any stationarity in space. Instead we allow the mean and covariance to vary spatially. In practice, we first remove a common trend from both $X$ and $Y$ to approximate stationarity in time. The trend is calculated as a global average at each time $t$ based on both random fields, thus the detrending has no effect on the spatially varying mean and the following test. Furthermore, we do not require $\{X_t(s)\}_{t=1}^N$ and $\{Y_t(s)\}_{t=1}^N$ to be Gaussian, though the

climate model data can be approximated by a Gaussian random field.

To compare the mean and covariance functions of two spatiotemporal random fields, we consider the following two hypotheses:

(i) $H_0 : \mu_X = \mu_Y$   vs.   $H_a : \mu_X \neq \mu_Y$,

(ii) $H_0 : C_X = C_Y$   vs.   $H_a : C_X \neq C_Y$.

The two test statistics for these two hypotheses are $TS1$ and $TS2$, which are explained in detail in the following two subsections. Because the empirical distributions of $TS1$ and $TS2$ have been derived under $H_0$, their p-values can be calculated. The p-values for these two hypotheses are ultimately what are used to evaluate the comparison between two fields, in this case between the known model field and a CFR. Another available test for evaluating the difference between two climate fields is to combine hypotheses (i) and (ii) into one single test, as in Li and Smerdon (2012) and Li et al. (2016). We omit this joint test

because the focus of this paper is to understand why the mean and covariance in a reconstructed field behave differently. Thus, each individual test is sufficient and more pertinent for such a purpose.

### 3.1.1 Mean comparison

The mean surface of a given climate field is a measure of its spatial variability across the global domain. In statistics, this is called the first moment of a spatiotemporal process and usually carries very important information about the distribution of the

270 random process. Comparisons between the mean structures between two climate fields is therefore fundamental for assessing their relative characteristics. The mean structure will be compared in subspaces that contain the major variability of the climate field, so we start by defining the subspaces and projected mean differences prior to defining the test statistics ($TS1$). We denote the $i$th eigenvalues and eigenfunctions, also called empirical orthogonal functions (EOFs), corresponding to $\hat{C}_X$ by $\{\hat{\lambda}_X^i\}$ and $\{\hat{\phi}_X^i\}$, where $\hat{C}_X$ denotes the sample covariance function using all time points. Then we define a sequence of vectors consisting

of the projected mean differences on the first $L$ eigenfunctions:

$$\hat{\psi}_k = (< \hat{\mu}_{X,k} - \hat{\mu}_{Y,k}, \hat{\phi}_X^1 > \cdots < \hat{\mu}_{X,k} - \hat{\mu}_{Y,k}, \hat{\phi}_X^L >)^T \tag{1}$$

for $1 \leq k \leq N$, where $< x, y >= x^T y$, and $\hat{\mu}_{X,k}$ ($\hat{\mu}_{Y,k}$) denotes the sample mean based on the recursive subsamples $\{X_t(s)\}_{t=1}^k$ ($\{Y_t(s)\}_{t=1}^k$). The reason we employ recursive subsamples is that we use a sampling strategy known as self normalization for time series data developed by Zhang and Shao (2015). Self normalization is an alternative to bootstrapping but has shown nice

properties when used for time series such as preserving temporal correlation and not requiring a tuning parameter. Recursive samples are obtained by drawing samples from time 1 to $k$ where $k = 2, ..., N$, meaning each time the new sample is formed by expanding the previous sample by adding the current observation. More details can be found in Zhang and Shao (2015).

Our test statistic for hypothesis (i) is therefore

$$TS1(L) = N\hat{\psi}_N^T V_\psi^{-1}(L)\hat{\psi}_N, \tag{2}$$

where $V_\psi = \frac{1}{N^2}\sum_{k=1}^{N}k^2(\hat{\psi}_k - \hat{\psi}_N)(\hat{\psi}_k - \hat{\psi}_N)^T$. The parameter $L$ is user chosen and determines how many eigenfunctions are to be used in the test.

### 3.1.2 Covariance comparison

The covariance structure refers to the correlation and the variance of climate observations over different locations. It is called second moment in statistics. When the climate field can be approximated by a Gaussian random field, the first and second moments determine the distribution of the entire random field. The covariance structure refers to either the local correlation or far-field correlation driven by so-called teleconnections within climate fields, and thus is an important description of the large-scale physical dynamics that underlie the climate system. To allow comparisons between leading patterns in modeled or reconstructed fields, we modify the test for covariance to make it suitable for comparing two cross-covariance functions. We again define subspaces and projected differences of a covariance structure. Let $C_X^{1,2}(s,s')$ and $C_Y^{1,2}(s,s')$ be the cross-covariance function for $s \in D_1$ and $s' \in D_2$ and let $\hat{C}_X^{1,2}(s,s')$ and $\hat{C}_Y^{1,2}(s,s')$ denote the sample cross-covariance function for $X_t(s)$ and $Y_t(s)$ based on all time points. We perform a SVD on $\hat{C}_X^{1,2}(s,s')$ or $\hat{C}_Y^{1,2}(s,s')$, say on $\hat{C}_X^{1,2}(s,s')$:

$$\hat{C}_X^{1,2}(s,s') = U'DV, \tag{3}$$

where $U$ and $V$ are orthogonal matrices with columns being $u_1,...,u_n$ and $v_1,...,v_m$ for $n$ and $m$ grid cells in subregion $D_1$ and $D_2$, respectively. The computational complexity for SVD is $O(min\{mn^2, m^2n\})$, and thus the computation can be an issue for a large $n$ and/or $m$. However, it is not a challenge here because the SVD is performed only on subregions $D_1$ and $D_2$. Let $\hat{C}_{X,k}^{1,2}(\hat{C}_{Y,k}^{1,2})$ denote the sample cross-covariance based on recursive subsamples $\{X_t(s)\}_{t=1}^k$ ($\{Y_t(s)\}_{t=1}^k$). That is $\hat{C}_{X,k}^{1,2}$ is the sample cross-covariance of $\{X_t(s_1)\}_{t=1}^k$ and $\{X_t(s_2)\}_{t=1}^k$. We define a sequence of matrices by the projected covariance differences, $C_k = [c_k^{i,j}]$, where $c_k^{i,j} = <\hat{u}_X^{iT}(\hat{C}_{X,k}^{1,2} - \hat{C}_{Y,k}^{1,2}), \hat{v}_X^j >, 1 \le k \le N$, $1 \le i, j \le L$, and $1 \le L \le min\{m,n\}$.

Let $\hat{\alpha}_k$ be the vectorized $\hat{C}_k$. The test statistic for hypothesis (ii) is

$$TS2(d) = N\hat{\alpha}_N^T V_\alpha^{-1}(d)\hat{\alpha}_N, \tag{4}$$

where $d$ is the length of the unique component in $\hat{\alpha}_k$, which contains the elements on and below the main diagonal of $C_k$. That is, $d = L(L+1)/2$ and $V_\alpha(d) = \frac{1}{N^2}\sum_{k=1}^{N}k^2(\hat{\alpha}_k - \hat{\alpha}_N)(\hat{\alpha}_k - \hat{\alpha}_{N_2})^T$, where $k^2$ is a weight to account for the sample size of the recursive subsample comprising observations from 1 to $\lfloor k/2 \rfloor$ in the estimation of $\alpha_k$. Again $L$ is chosen by the user and can be determined by the cumulative percentage of total variation.

Additionally, the test statistics of above two tests will change if we calculate the sample covariance matrix based on $Y$ process rather than the $X$ process, because the EOFs from $Y$ are different than those from $X$. Thus, they are not exchanagable but we have fixed the sample covariance matrix based on the $X$ process because the goal of our application is to evaluate the skill of CFRs by comparing them to their known targets, the climate model output on which each PPE is based.

## 4   Results

### 4.1   Mean-structure skill

Despite the formalism of the preceding section, the important implication is that comparisons between modeled and reconstructed fields can be measured in terms of p-values based on a null hypothesis that similarities are within the range of comparisons between two random spatiotemporal fields. A p-value close to 0 indicates the difference between modeled and reconstructed fields is statistically significant against this null hypothesis, while p-values close to 1 indicates the difference could be explained by random chance. In other words, p-values close to 0 indicate poor reconstructions, while p-values close to 1 indicate a CFR and target field are statistically similar within the range of random chance. Moreover, these comparisons are broken out among the leading spatiotemporal patterns in each field. Investigating the large-scale spatial patterns is an effective way to evaluate the skill of the CFRs as the important features of the spatiotemporal fields are often masked by substantial noise. Therefore, while such comparisons can be done for any number of leading principal components, we focus herein on the leading five in each field as these 5 EOFs consist of more than 80% of the total variability and largely represent the dominant spatial patterns of random fields. Subsequent comparisons are made between the CFRs in each model-based PPE and the known model field during the reconstruction interval (850-1849 CE).

The mean-structure performance, in terms of the developed skill metric for the five leading EOFs, is shown for each CFR method within each of the model-based PPEs in Figure 2; results are shown for PPEs using pseudoproxies with $SNRs = 0.5$ (upper panel) and $SNRs = \infty$ (bottom panel). Several general observations associated with Figure 2 stand out as consistent with previous work using traditional skill measures in Smerdon et al. (2016). First, there are clearly differences across each of the model-based PPEs indicating that CFR performance depends strongly on the spatiotemporal characteristics of the underlying model fields. Consistent with Smerdon et al. (2016), the methods perform best in the CCSM-, GISS-, and MPI-based PPEs, while the BCC and IPSL models appear to present the most challenging tests for the CFR methods. Secondly, the level of noise in the pseudoproxies has an important and expected impact on the nature of the methodological performance. Particularly for the CCSM-, GISS- and MPI-based PPEs, the no-noise experiments yield much higher skill scores than the $SNR = 0.5$ experiment. Notably, however, even the no-noise PPEs yield CFRs with variable skill that depends on method and model.

With regard to the performance of specific methods, TTLS and TTLH are generally most skillful across the top five EOFs in the CCSM, GISS, and MPI PPEs, although that is not true across all of the EOFs and is more ambiguous for the CCSM experiment with $SNR = 0.5$. It is also true that the TTLS and TTLH methods perform similarly within each model-based PPE across the top five EOFs, which is not surprising given the close methodological lineage of the two methods (Smerdon et al., 2016). Similarly, the CCA and RIDGE methods have similar skill performance for each of the five EOFs across the PPEs, although the CCSM experiment shows some ambiguity with regard to these general observations again in the $SNR = 0.5$ case. Finally, the methods collectively perform the worst within the BCC and IPSL PPEs, a finding that is again consistent with the mean bias assessment in Smerdon et al. (2016) who found the largest mean biases in the BCC and IPSL PPEs.

In addition to the above general observations, the applied skill metric allows the skill associated with each of the leading EOFs to be separated. Nothing similar to these separations were performed in Smerdon et al. (2016) and they indicate a

complicated structure associated with skill across each of the model-based PPEs and tied to the applied method. For instance, some methods perform very well on several leading EOFs, while performing very poorly on several others (CCA in the CCSM

PPE or CCA and RIDGE in the GISS PPE). Other methods perform poorly on the leading EOF, while performing very well on the remaining EOFs (TTLS and TTLH in the CCSM and MPI PPEs) or perform poorly on all EOFs except the 5th EOF (CCA, Ridge, and TTLH in the IPSL PPE). The implication of these assessments is that there is a rich structure to the performance of the methods across the different model-based PPEs, but the reasons for this performance is not immediately obvious from these assessments. We therefore perform a similar analysis in the next subsection for the covariance-structure skill, before working

to more deeply understand the performance of the CFR methods as indicated by the applied skill metric.

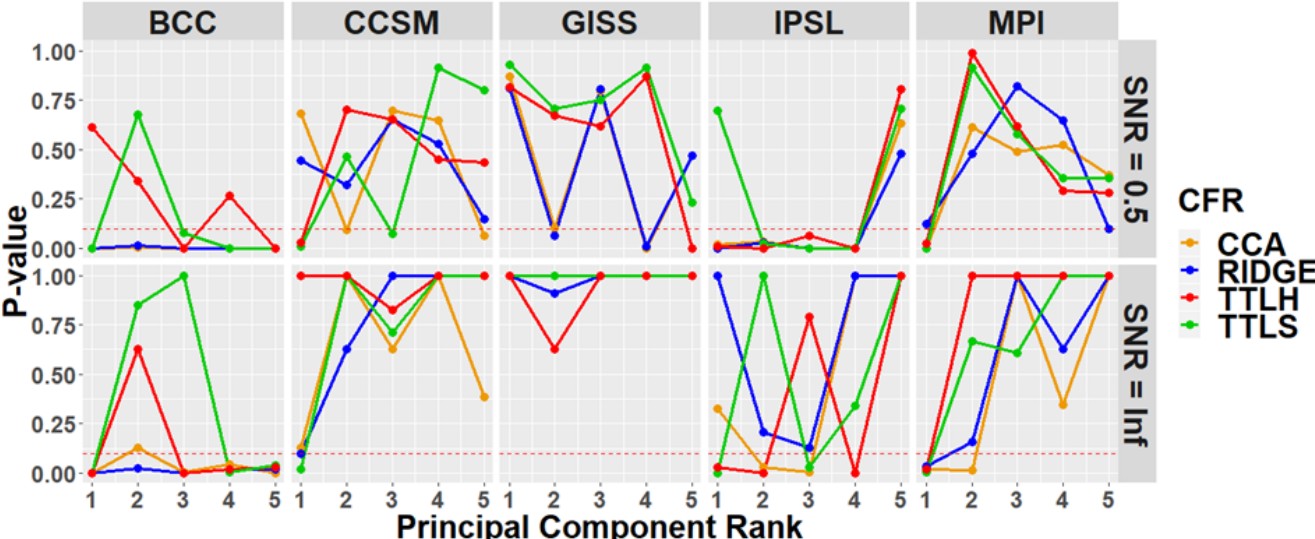

**Figure 2. CFR Mean-structure performance within each of the model-based PPEs.** Derived p-values are shown for the mean comparison between the target model and the CFRs based on the described skill metric and presented for the leading five principle components. Upper and bottom panels show skill assessment for the $SNR = 0.5$ and $SNR = \infty$ PPEs, respectively. PCs with p-values greater than the significance level of 0.05 (dotted line) are considered to skillfully recover the mean structure of each model.

### 4.2 Covariance-structure skill

Similar to the mean-structure comparison, we employ the applied skill metric to evaluate how derived CFRs reproduce the known covariance of the climate model simulations. We first note, however, that the covariance comparisons between the CFRs and the known climate model fields over the entire reconstruction domain yielded results that were universally unskillful. In

other words, our analyses yielded p-values equal or close to 0 for all methods at all five leading EOFs and across all PPEs. This result is perhaps not unexpected given an established understanding that there are large regions with very low skill throughout the global CFR domain. Smerdon et al. (2016) demonstrated that many regions of the reconstructed fields have small and insignificant correlations relative to the known model fields, while locations among the tropics and over dense pseudoproxy

sampling locations achieve much larger correlations. These collective results thus suggest that if the global domain is used to

identify EOFs many of the locations will be defined by variability dominated by noise. Alternatively, constrained domains that encompass dominant regions of variability can be used to target leading EOFs that are less susceptible to noise. We therefore modify our approach in this section to describe comparisons between areas of dominant teleconnections in the model fields.

     Our modified approach is to analyze the covariance structure only in regions where the teleconnection associated with the El Niño-Southern Oscillation (ENSO) is dominant. We specifically focus on ENSO because it is the leading mode of

internal variability on a global scale, making it easy to identify and likely strongly expressed in the leading few modes of each climate model simulation. We examine the ENSO dependencies by computing the correlation between the time series of averaged temperatures over the Niño3 region (5°N-5°S, 150°W-90°W), and the time series at all other grid points in the global temperature field. Maps of these correlations for each climate model are shown in Figure 3. We discard locations that are proximal to the ENSO region (local covariance structure) that are not the consequence of the ENSO teleconnection (large-scale

covariance structure). An empirical covariance estimate suggests that pairs within 10,000 km are due to this proximity. Thus, we exclude the locations in the orange shaded area (20°S-20°N, 150°E-35°W) that are within 10,000 km from the center of the Niño3 region, as shown in Figure 3. After excluding these proximal locations, we choose the grid points that have significant positive or negative correlations with the Niño3 index in each model at the 10% significance level, which we interpret as reflecting each model's ENSO teleconnection pattern. Because the selected grid points vary for different climate models, we

use the collection of overlapping grid points (black dots in Figure 3) from all five climate models.

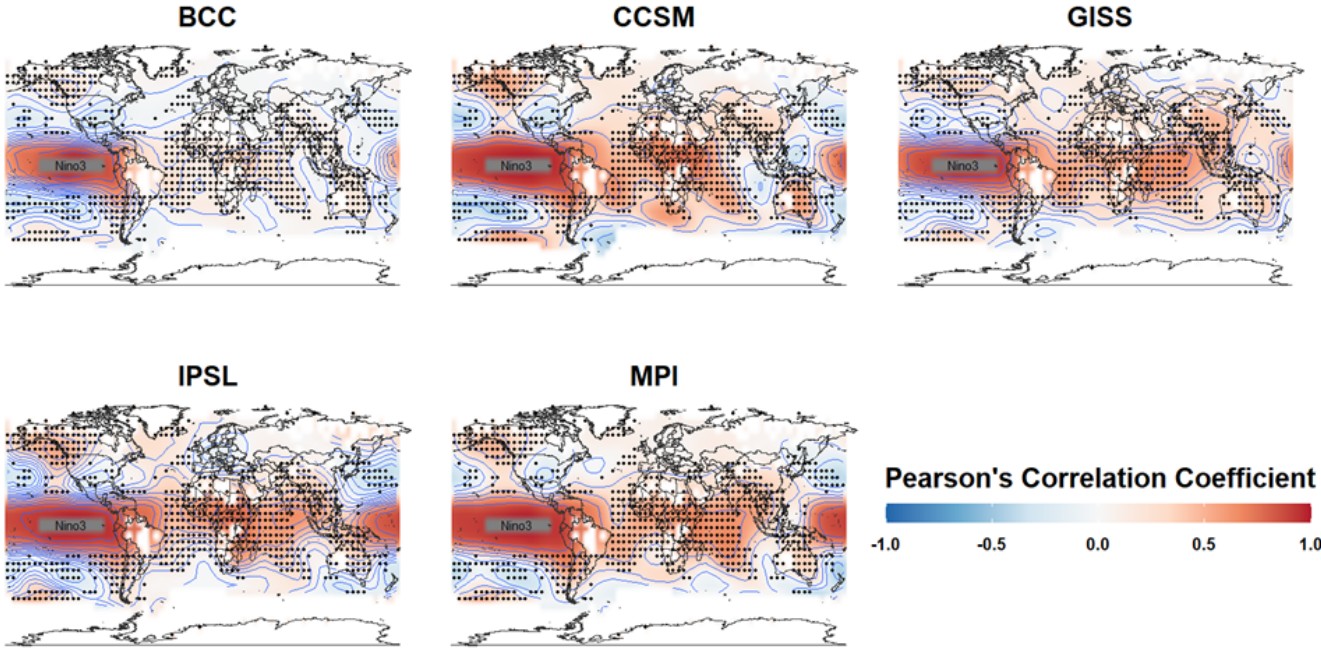

**Figure 3. Model-based correlations between the Niño3 index and temperatures at all other grid points.** The Niño3 index is computed as the average sea surface temperature within the indicated box in the tropical Pacific (5N-5S, 150W-90W). Global correlations are indicated by the color scaling. Shared grid-point locations where local temperatures significantly correlate with the Niño3 index (p < 0.1) are indicated by the same black dots in each panel ($D_2$). The orange shaded region in each map indicates where locations have been excluded to focus predominantly on far-field teleconnection structure.

The p-values for the modified covariance-structure skill metric assess how well the large-scale teleconnection patterns associated with ENSO are reproduced in the CFRs, relative to the known model fields. These results are shown in Figure 4, which presents the p-values for all four CFRs for the leading five PCs using the $SNR = 0.5$ and $SNR = \infty$ PPEs. Relative to assessments over the entire domain, stronger associations between the CFRs and known model fields are observed when the covariance structure is limited to the ENSO teleconnection regions. Even with a constrained focus on the ENSO teleconnection regions, however, the covariance-structure skill is still limited across most of the methods and model-based PPEs. The TTLS and TTLH methods are again the most skillful across all of the methods. In the $SNR = \infty$ case, skill is detected for the CCA method across all model-based PPEs and there is some skill for the RIDGE method except for the IPSL and MPI PPEs. Interestingly, the skill of TTLS and TTLH is higher for most of the EOF patterns in the $SNR = 0.5$ PPEs based on the CCSM, GISS, and IPSL models, relative to the no-noise case, while it is more typical to have skill reduction for $SNR = 0.5$ as in the CFRs derived for the BCC and MPI PPEs. Specifically, compared to the no noise case, CCA and RIDGE results show skill reduction for $SNR = 0.5$ for all PPEs. We also note that for the covariance comparison, it is particularly important to examine the skill of CFRs in the first PC because the first EOF contains over 80% of the total variation in all models (see

further discussion in Section 4.4), except for BCC (the first EOF contains approximately 45% of the variability). TTLS and
TTLH within the CCSM PPE are most skillful over the first two PCs showing evident skill in the subspace containing over 90%
of the total variation. While most of the model-based PPEs indicate some skill associated with at least the TTLS and TTLH
methods, the BCC and MPI-based PPEs stand out as yielding almost no skill at any PC for all the methods in the $SNR = 0.5$
PPEs. In the case of no noise, CCA, TTLS, and TTLH have skill associated with over 80% of the variation in MPI, while CCA
and RIDGE show very little skill in all EOFs other than the first.

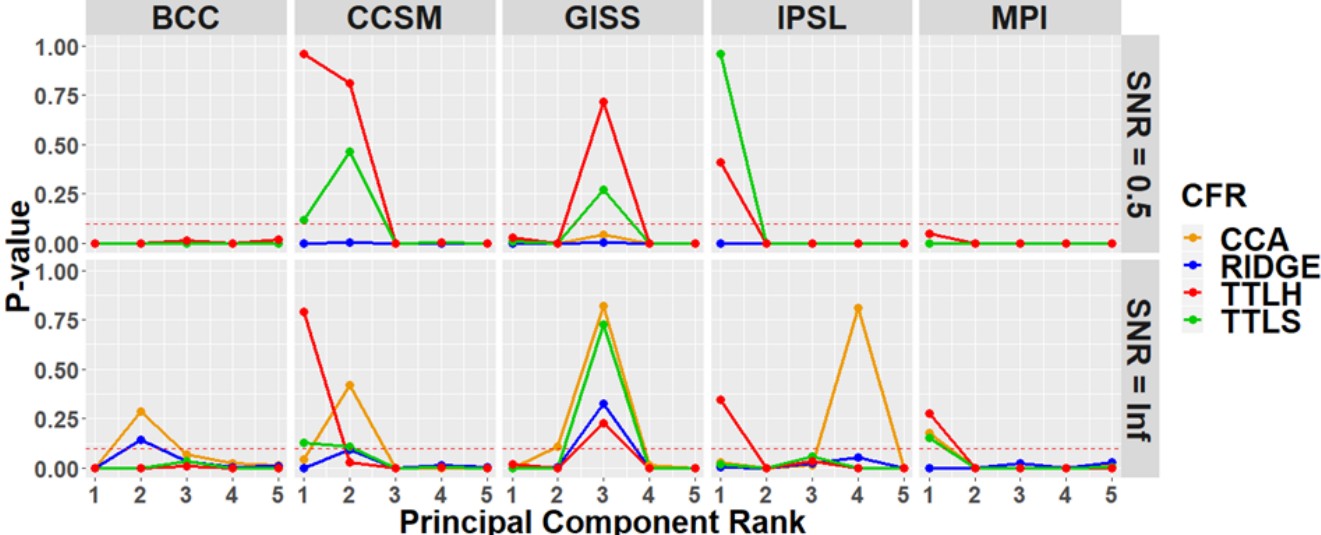

**Figure 4. CFR covariance-structure performance within each of the model-based PPEs in ENSO teleconnection regions only.** Derived
p-values are shown for the covariance comparison between the target model and the CFRs based on the described skill metric and presented
for the leading five principle components. Comparisons are performed only in the ENSO teleconnection regions shown in Figure 3 ($D_2$
locations). Upper and bottom panels show skill assessment for the $SNR = 0.5$ and $SNR = \infty$ PPEs, respectively. PCs with p-values
greater than the significance level of 0.05 (dotted line) are considered to skillfully recover the covariance structure in the strong ENSO
teleconnection regions.

To complement the analysis of the covariance-structure skill in the ENSO teleconnected regions, we investigate the
proportion of variance explained by the first five leading EOFs of the ENSO teleconnection dominant region ($D_2$). Figure 5
shows that more than 30% of the variance is explained by the first EOF in CCSM and IPSL models, while the other three models
present less than 30% of the variance in their first EOFs. This feature is likely linked to the results in Figure 4 ($SNR = 0.5$)
showing that the first PC of BCC, GISS, and MPI models fail to show skill. This signal is especially weakly expressed in the
leading modes of the modeled data and not well represented in its CFRs when $SNR = 0.5$ (only TTLH showing p-value of
0.03), although Figure 3 shows that MPI exhibits a strong teleconnection signal. This is particularly so as noise is added to
the pseudoproxies and especially for the CCA and RIDGE methods. Thus for the MPI model, the skill of most of the CFRs is
associated with the first PC when there is no noise ($SNR = \infty$) but none when the noise is present ($SNR = 0.5$).

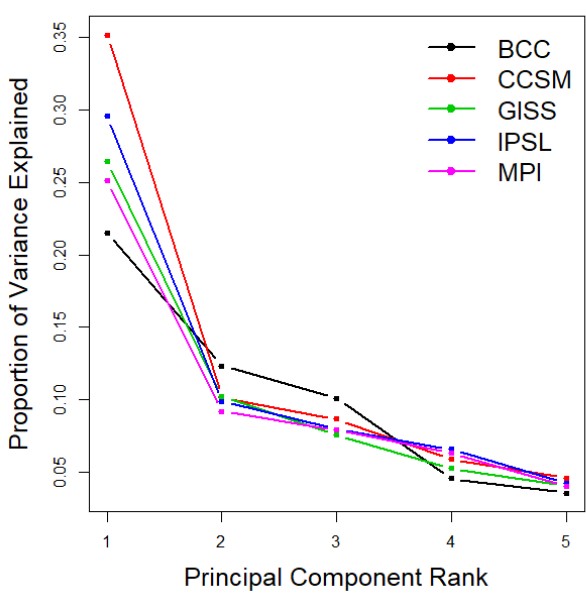

**Figure 5. Eigenvalue spectra of ENSO-teleconnected regions for each of the last millennium simulations:** The spectra of the ENSO-teleconnected regions for each last-millennium simulation are computed as the ratio of between the first five eigenvalues and the cumulative sum of all eigenvalues over the ENSO dominant region ($D_2$).

### 4.3 Cumulative CFR Skill

Figures 2 and 4 present the performance of the CFRs over the first five EOFs in each of the PPEs, but these comparisons do not characterize how the skill accumulates over the collection of EOFs and how much total variability in the field in represented in the skill assessment. Figures 6 and 7 therefore present the p-values for the overall skill of CFRs associated with the mean- and covariance structure for the $SNR = 0.5$ PPEs, respectively, but in this case they are derived according to the proportion of the cumulative variability explained by a successive number of leading PCs. Despite the indication of skill across multiple PCs

demonstrated in Figures 2 and 4, the skill as a function of cumulative variance reveals that most methods across most PPEs do not recover mean-structure skill beyond about 30% of the total variability. Regarding mean-structure skill specifically, TTLH exhibits the most skill within the CCSM and GISS PPEs, and only up to about 20% - 30% of the total variation. In the GISS model, unlike the other models, all of the CFRs except RIDGE present skill up to 20% of the total variability. On the other hand, CFRs in the BCC and MPI PPEs show no skill for cumulative EOFs. The mean comparison results for the no-noise cases

also exhibit very similar results (results not shown).

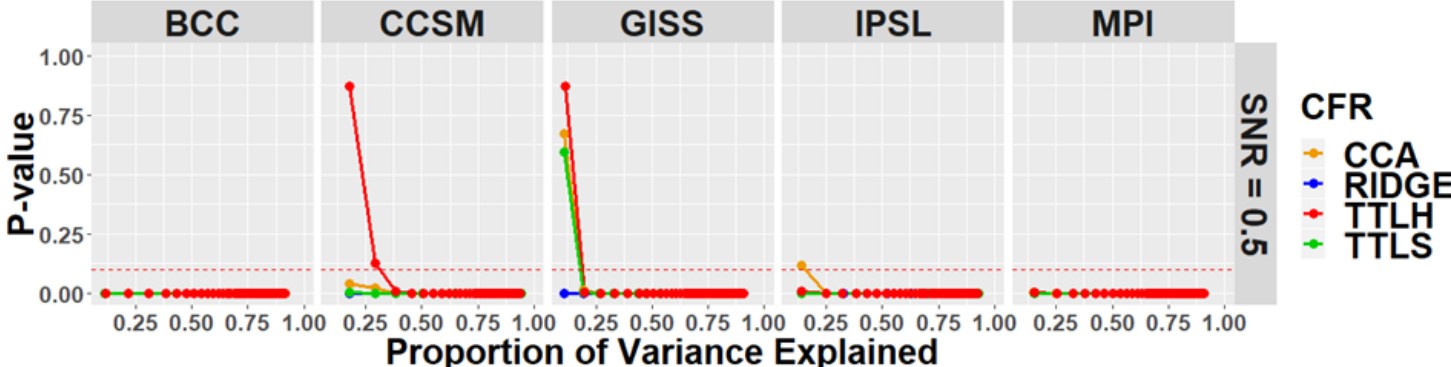

**Figure 6. Mean comparison of the successive order of principal components during the reconstruction period:** For every climate model, p-values for the mean comparison between the target model and the CFRs based on the described skill assessment are presented for the successive range of PCs. PCs with p-values greater than the significance level of 0.05 (dotted line) are considered to skillfully recover the mean structure.

Regarding the cumulative covariance-structure skill in the ENSO teleconnection regions shown in the top panel of Figure 7, only TTLS and TTLH in the CCSM and IPSL PPEs show skill. Because the first PC of these CFRs in CCSM and IPSL already consists of more than 80% of the total variation, TTLS and TTLH are very skillful in recovering the teleconnection pattern in the CCSM and IPSL PPEs. There is also some skill detected for TTLS and TTLH in the GISS PPE (Figure 4), but because the percent of variation in the 3rd PC is very small (less than 10% of the total variation) the skill at this PC is masked by the variation of the previous two PCs. Moreover, in the case of no noise, CFRs of the BCC and GISS PPEs consistently show no skill and there is some skill associated with TTLS and TTLH at the first PC in CCSM, IPSL, and MPI PPEs in the bottom panel of Figure 7.

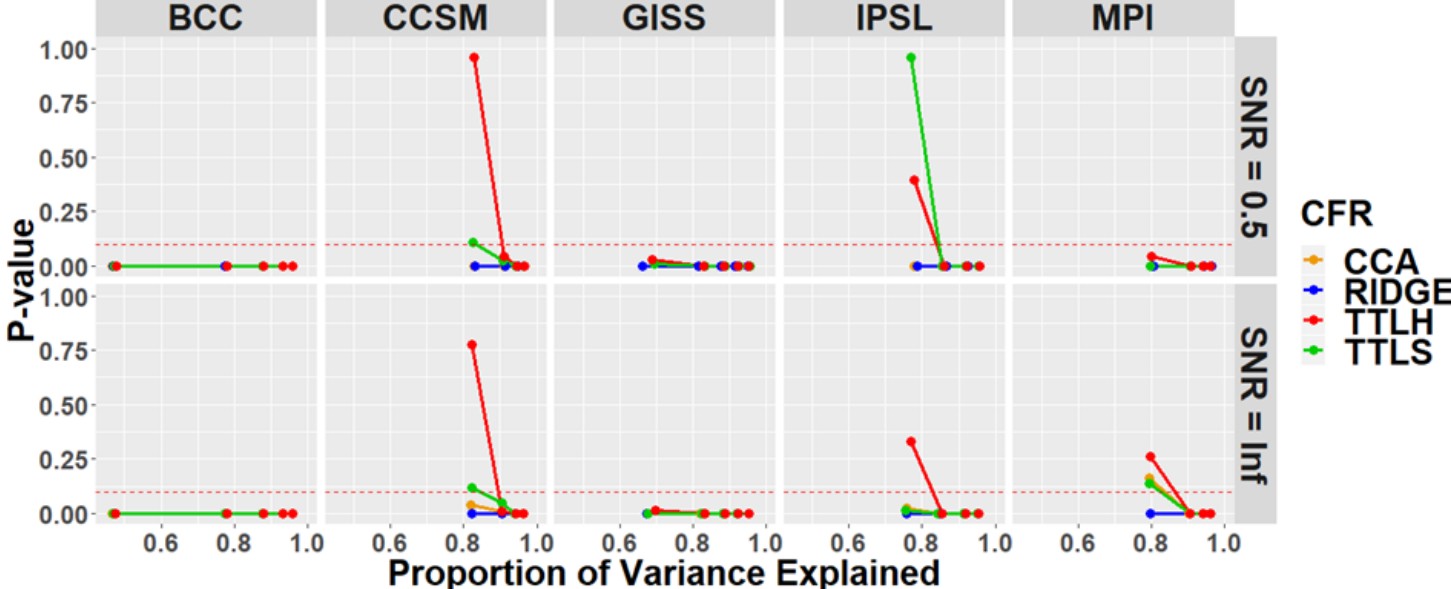

**Figure 7. Covariance comparison over the successive order of principal components within only ENSO-teleconnection regions:** For every climate model, p-values are derived for covariance comparisons only in the $D_2$ locations of the target model and the CFRs using the described skill assessment for the successive range of PCs based on $SNR = 0.5$ and $SNR = \infty$ PPEs shown in top and bottom panels, respectively. PCs with p-values greater than the significance level of 0.05 (dotted line) are considered to have some skill in recovering the ENSO teleconnection structure.

Figure 6 - 7 indicate that even though CFRs show some skill for each individual PC, the cumulative variability that is skillfully explained must be evaluated for a more complete picture of methodological performance. This is especially true for the TTLS and TTLH methods. Despite showing outstanding skill recovering the modeled mean structure at each PC in most of the climate models in Figure 2, Figure 6 shows that the two methods often do not account for skill up to higher order cumulative EOFs. For example, both TTLS or TTLH are skillful only up to 20-30% of the total variation of the climate field. Additionally, we note that the CCA method poorly recovers the mean structure in most of the climate models and both CCA and RIDGE are poor in recovering the covariance structure in all five climate models when the comparison is projected onto the cumulative EOF basis function.

## 4.4 Interpreting the Mean and Covariance Skill Assessments

While the preceding subsections provided some guidance regarding the performance and comparisons of the CFR methods in the multiple model-based PPEs, it is still unclear why the methods perform differently and how they depend on different characteristics of the climate simulated by each model. In the following subsection, we therefore characterize the features of the temperature fields simulated by the models and the underlying consequences for the various CFR methods. We interpret the skill assessments by exploring several features of the CFRs and the underlying model fields on which the PPEs are based:

(i) the percent variance explained by the leading EOFs in the modeled temperature field, (ii) the temporal stability of the EOF structure in the reconstruction and calibration periods, and (iii) the degree to which the spatiotemporal variability in the modeled temperature fields are represented by the locations where pseudoproxies are sampled.

### 4.4.1 Structure of the Eigenvalue Spectrum

Because each of the CFR methods investigated in this study are forms of regularized multivariate regression, they all share a similar feature, namely they each only target a few of the leading EOFs in the target temperature field. An important control on the skill of CFRs is therefore tied to how much of the variance in the target temperature field is explained by the leading EOFs. We therefore hypothesize that the PPEs based on the climate model simulations with significant amounts of variance in a few leading EOFs will be those experiments in which the CFRs perform most skillfully.

In Figure 8, the variance explained by the first five EOF-PC pairs (same-rank pair of an empirical orthogonal function and its principal component) in each model is represented as the ratio between each of the five eigenvalues of the decomposition of the covariance matrix of the surface temperature fields and the sum of all the eigenvalues. These calculations indicate that except for the BCC model, a large portion of the variance is explained by the leading EOF-PC pairs in each of the modeled surface temperature fields. Additionally, the proportion of the explained variance in the first eigenvalue is relatively high in all of the models except for BCC. Based on these results alone, the BCC model would be predicted to form the basis for the most difficult PPE, an expectation that is largely reflected in the skill assessments from Figures 2 and 4.

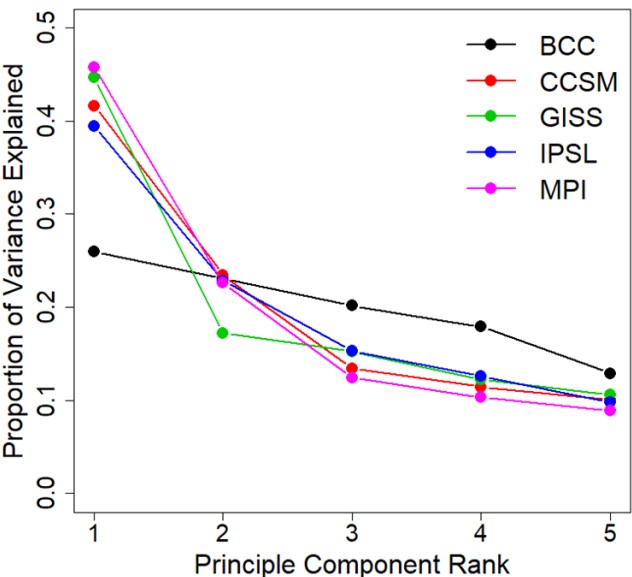

**Figure 8. Eigenvalue spectra for each of the last-millennium simulations:** The spectra for each last-millennium simulation are computed as the ratio between the first five eigenvalues and the cumulative sum of all eigenvalues.

In addition to providing a broad assessment of the relative challenges presented by the individual model-based PPEs, the eigenvalue spectra for each of the CFRs in each of the model experiments also indicate that the similarity between the variance explained in the first several EOFs of the target and reconstructed fields is largely indicative of the performance of the individual CFR methods. In particular, the proportions of the first eigenvalues in the TTLS and TTLH CFRs are almost equivalent to those of the true model fields from the CCSM, GISS, IPSL, and MPI simulations (Figure 9). This is reflective of the fact that those two methods generally performed the most skillfully in both of the skill assessment metrics. In contrast, the proportions of the first eigenvalues of the CFRs in the BCC PPE are significantly lower than that of the true model, which matches with the relatively poor skill of all CFR methods based on the BCC PPE. CFR performance is therefore strongly associated with how well the first EOF represents the total variation in the targeted climate field and how well that variance is reproduced in a given CFR.

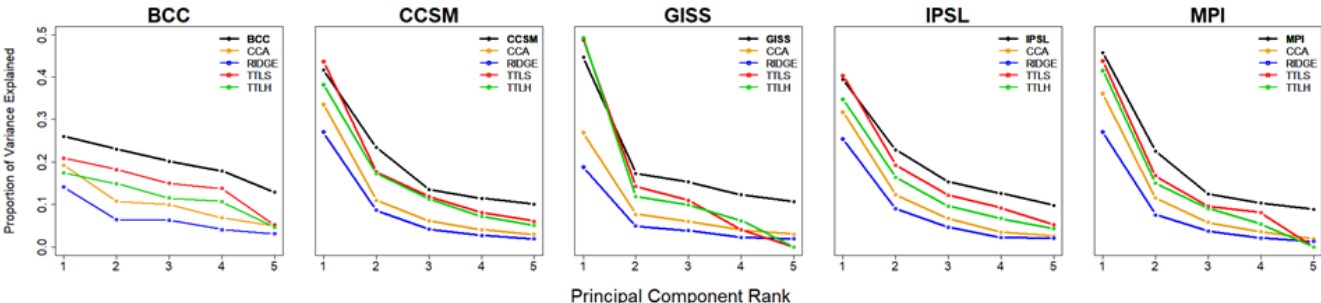

**Figure 9. Eigenvalue spectra for each last-millennium simulation and the CFRs for each psuedoproxy experiment:** The spectra for each last millennium simulation and associated CFR are computed as the ratio between the first five eigenvalues and the cumulative sum of all eigenvalues.

While the above analyses of the eigenvalue spectra give important insights into the difficulty of reconstructing a given climate field and the likely performance of a reconstruction that targets such a field, the variance explained by a given set of EOF-PC pairs alone may not be fully indicative of reconstruction performance. For instance, it is possible that the EOFs in the reconstruction are reordered so that they do not represent well the spatial characteristics of any given EOF in the target field. It is therefore useful to assess how well the spatial characteristics of specific EOFs in a CFR represent the spatial characteristics of the EOFs in a targeted field.

To assess this feature and allow for the fact that a given reconstructed EOF may be ordered differently than the equivalent EOF in the target field, we take the inner product between each of the first three EOFs in the reconstructed and targeted fields (this is similar to the spatial correlation statistic often discussed in the climate literature, e.g. Coats et al., 2013; Baek et al., 2017). If the absolute value of the inner product is close to 0, it suggests that the spatial patterns represented by two EOFs are very different, while if the inner product is close to 1, it implies that they are equivalent. The p-values testing the significance of the inner products can be derived using bootstrap analysis. Each sample is obtained by bootstrapping spatial locations at each

time point, from which the inner product of the CFR and the associated true climate model is calculated based on the sampling. For each inner product pair, we perform bootstrapping 1,000 times and calculate the p-value of the observed inner product.

Table 1 presents the inner products of the first 3 EOFs for the CFRs and the corresponding climate model fields, with significance also indicated. Inner products along the diagonals that are close to one and marked as significant indicate that the order of their corresponding EOFs together with their spatial representations in the CFRs are similar to those of the targeted climate field. Values close to one, significant, and off the diagonal would indicate a potential reordering of the reconstructed EOFs, relative to the EOF structure of the target field. Collectively, the inner products indicate that in addition to reflecting similar eigenvalue spectra (Figure 9), CFRs targeting the CCSM and MPI models also produce EOFs that are similarly ordered with patterns that well represent the true model EOF patterns. This is represented by high and significant inner product values along the diagonals; the opposite is true of most experiments with the BCC model. In summary, in order for the CFRs to well depict the mean and covariance structure of the true climate model, the first few leading EOFs should carry the majority of the total variation while also capturing the spatial features of the targeted EOFs as shown in Table 1.

**Table 1.** EOF inner product of the true model fields and the associated CFRs

| | | EOF of CFRs | | | | | | | | | | | |
| --- | --- | --- | --- | --- | --- | --- | --- | --- | --- | --- | --- | --- | --- |
| | | CCA | | | RIDGE | | | TTLS | | | TTLH | | |
| | | 1 | 2 | 3 | 1 | 2 | 3 | 1 | 2 | 3 | 1 | 2 | 3 |
| BCC | 1 | 0.814† | 0.350 | 0.387 | 0.772† | 0.006 | 0.594 | 0.263 | 0.312 | 0.752‡ | 0.588 | 0.511 | 0.256 |
| | 2 | 0.225 | 0.548 | 0.328 | 0.172 | 0.505 | 0.056 | 0.104 | 0.669† | 0.199 | 0.230 | 0.400 | 0.569 |
| | 3 | 0.108 | 0.665 | 0.153 | 0.078 | 0.735 | 0.209 | 0.599 | 0.051 | 0.410 | 0.476 | 0.347 | 0.412 |
| CCSM | 1 | 0.967‡ | 0.163 | 0.006 | 0.966‡ | 0.179 | 0.028 | 0.955‡ | 0.191 | 0.039 | 0.961‡ | 0.078 | 0.138 |
| | 2 | 0.122 | 0.937‡ | 0.115 | 0.141 | 0.908‡ | 0.197 | 0.085 | 0.809† | 0.302 | 0.001 | 0.857† | 0.238 |
| | 3 | 0.050 | 0.166 | 0.822 | 0.037 | 0.226 | 0.741 | 0.144 | 0.366 | 0.681 | 0.180 | 0.396 | 0.689 |
| GISS | 1 | 0.906‡ | 0.046 | 0.007 | 0.914‡ | 0.006 | 0.005 | 0.855† | 0.082 | 0.164 | 0.809† | 0.165 | 0.268 |
| | 2 | 0.113 | 0.914‡ | 0.010 | 0.059 | 0.917‡ | 0.057 | 0.269 | 0.217 | 0.754‡ | 0.272 | 0.290 | 0.733† |
| | 3 | 0.027 | 0.034 | 0.773† | 0.014 | 0.106 | 0.767† | 0.015 | 0.657 | 0.221 | 0.165 | 0.608 | 0.227 |
| IPSL | 1 | 0.932‡ | 0.261 | 0.116 | 0.911‡ | 0.344 | 0.087 | 0.949‡ | 0.106 | 0.164 | 0.969‡ | 0.080 | 0.052 |
| | 2 | 0.188 | 0.822† | 0.350 | 0.249 | 0.747 | 0.431 | 0.042 | 0.838† | 0.176 | 0.070 | 0.857† | 0.136 |
| | 3 | 0.133 | 0.426 | 0.503 | 0.176 | 0.487 | 0.465 | 0.080 | 0.428 | 0.443 | 0.022 | 0.407 | 0.548 |
| MPI | 1 | 0.972‡ | 0.116 | 0.111 | 0.959‡ | 0.177 | 0.128 | 0.982‡ | 0.061 | 0.065 | 0.982‡ | 0.071 | 0.034 |
| | 2 | 0.123 | 0.953‡ | 0.057 | 0.193 | 0.929‡ | 0.061 | 0.054 | 0.943‡ | 0.002 | 0.079 | 0.944‡ | 0.008 |
| | 3 | 0.088 | 0.003 | 0.861† | 0.096 | 0.017 | 0.852† | 0.043 | 0.078 | 0.846† | 0.040 | 0.054 | 0.917‡ |

Note: Significances of inner products are denoted by ‡ and † for the 10% and 20 % levels respectively.

### 4.4.2 Temporal stability of the leading EOFs

An important underlying assumption of linear regression based CFR methods is that the identified patterns in the calibration period remain temporally stable back in time over the period of reconstruction. In particular, temporal stability refers to how much the leading patterns of modeled data in the reconstruction period and in the calibration period share in common and to what extent the order of leading patterns in the calibration period is preserved in the reconstruction period. If these patterns are not temporally stable, a key assumption of the reconstruction approach is violated and the skill of the reconstruction will be affected. Differences in the performance of CFR methods, such as the differences in the mean structures assessed in Figure 2, may therefore be explained by differences in the temporal stability of leading EOFs in the calibration and reconstruction intervals within each of the model simulations that form the basis of the PPEs. In other words, if the EOF character and structure within the reconstruction interval is similar to that of the calibration interval within a model simulation, all of the CFRs based on this model simulation are expected to capture the mean structure better than the CFRs based on simulations for which this is not the case. Despite the above significance for reconstruction methods, it is unknown whether teleconnections in the real climate system remain stable over centennial to millennial time scales or how widely they have varied if they are not stable (e.g. Coats et al., 2013).

To test the stability of the teleconnections in the model simulations, we again use the inner product as a measure of the similarity between spatial patterns, in this case between the EOFs in the calibration and reconstruction periods. These inner products are listed in Table 2 and the p-values of the inner products are again computed through bootstrapping; the pairs of EOFs that are significantly aligned are marked. If the inner product matrix in Table 2 contains the highest values along the diagonal and those values are significant in the bootstrapping experiments, it suggests that the order and character of the EOFs are similar in the calibration and reconstruction intervals. This is predominantly the case for the CCSM and MPI simulations, implying that the reconstruction period is defined by the same dominant pattern of leading EOFs in the calibration period. Moreover, for those two climate models, the order of the modes are preserved as well. In contrast, the BCC model reveals very weak associations between the calibration and the reconstruction periods, and IPSL only displays strong association for the first EOF.

The temporal stability assessment, when joined by the previous assessment of the eigenvalue spectra allow a more specific criterion for CFR methodological success: if a large fraction of the variability in the climate field is represented by a few leading EOFs, and the EOFs are stable across the calibration and reconstruction periods, the CFRs tend to recover the true mean structure well. Because BCC and IPSL simulations violate either or both of these two conditions, CFRs based on BCC and IPSL have reduced skill in this sense. Again the performance of TTLS and TTLH largely depends on how well the first few EOFs of the reconstruction period represent the dominant EOF patterns in the calibration period. On the other hand, CCA and RIDGE usually outperform the other methods when the reconstruction and calibration period share the total variation across a larger number of the leading EOFs. As an example, CCA and RIDGE well recover the mean structure in the CCSM and MPI PPEs because strong and distinct patterns are shared in all five leading EOFs of these model simulations. In contrast, CCA and

RIDGE do not perform well in the BCC PPE (Figure 2) because the BCC simulation carries less of the total variability in its leading modes, which are also not temporally stable between the calibration and reconstruction intervals.

**Table 2.** Inner product of EOFs derived in the calibration and reconstruction periods

| Inner Product | | EOFs of Calibration period | | | | |
|---|---|---|---|---|---|---|
| | | 1 | 2 | 3 | 4 | 5 |
| | 1 | 0.55 | $0.761^{\ddagger}$ | 0.02 | 0.207 | 0.167 |
| EOFs | 2 | 0.042 | 0.084 | 0.436 | $0.717^{\ddagger}$ | 0.389 |
| of | 3 | 0.16 | 0.334 | $0.605^{\dagger}$ | 0.555 | 0.253 |
| BCC (Recon) | 4 | 0.242 | 0.352 | 0.569 | 0.019 | 0.49 |
| | 5 | 0.63 | 0.327 | 0.128 | 0.216 | 0.45 |
| | 1 | $0.966^{\ddagger}$ | 0.093 | 0.057 | 0.034 | 0.037 |
| EOFs | 2 | 0.002 | $0.872^{\ddagger}$ | 0.28 | 0.091 | 0.084 |
| of | 3 | 0.098 | 0.32 | $0.82^{\dagger}$ | 0.289 | 0.173 |
| CCSM (Recon) | 4 | 0.017 | 0.026 | 0.335 | $0.804^{\dagger}$ | 0.273 |
| | 5 | 0.078 | 0.192 | 0.139 | 0.255 | $0.815^{\dagger}$ |
| | 1 | $0.861^{\ddagger}$ | 0.164 | 0.027 | 0.22 | 0.08 |
| EOFs | 2 | 0.194 | $0.895^{\ddagger}$ | 0.213 | 0.045 | 0.057 |
| of | 3 | 0.098 | 0.071 | 0.225 | $0.68^{\dagger}$ | 0.155 |
| GISS (Recon) | 4 | 0.247 | 0.053 | 0.57 | 0.272 | 0.196 |
| | 5 | 0.04 | 0.253 | $0.661^{\dagger}$ | 0.089 | 0.345 |
| | 1 | $0.886^{\ddagger}$ | 0.378 | 0.096 | 0.137 | 0.022 |
| EOFs | 2 | 0.223 | $0.643^{\dagger}$ | $0.627^{\dagger}$ | 0.162 | 0.062 |
| of | 3 | 0.222 | 0.616 | 0.496 | 0.439 | 0.087 |
| IPSL (Recon) | 4 | 0.111 | 0.051 | 0.455 | 0.573 | 0.329 |
| | 5 | 0.139 | 0.11 | 0.118 | 0.488 | 0.024 |
| | 1 | $0.976^{\ddagger}$ | 0.084 | 0.092 | 0.069 | 0.066 |
| EOFs | 2 | 0.077 | $0.943^{\ddagger}$ | 0.179 | 0.101 | 0.127 |
| of | 3 | 0.072 | 0.227 | $0.844^{\ddagger}$ | 0.352 | 0.06 |
| MPI (Recon) | 4 | 0.067 | 0.041 | 0.323 | $0.759^{\dagger}$ | 0.185 |
| | 5 | 0.048 | 0.076 | 0.075 | 0.145 | $0.724^{\dagger}$ |

Note: Significances of inner products are denoted by ‡ and †, for the 10% and 20% levels respectively.

### 4.4.3 Sampling locations

The sampling locations of proxies also play a key role in the performance of CFRs, because all CFR methods train their sta-
tistical models based on how the entire climate field relates to the climate variability reflected in proxy locations. If the climate
variability at sampling locations poorly represents the variability of the entire climate field, then it will be very challenging
for CFRs to reproduce the mean or covariance structure of the targeted climate. To investigate this possible issue, we sample
the climate from only the proxy sampling locations and then study the capacity of the climate at those locations to recover
the climate globally. This is carried out by directly using the EOFs at the sampling locations to estimate the climate at other
locations and examine the mean squared error (MSE) of the estimates.

In order to account for spatial correlation in this context, we first decorrelate the spatial climate simulation before fitting a
linear model and then add the correlation back after we obtain the estimates. More specifically, let $X_m^*(s,t)$ denote the spatially
decorrelated climate simulation obtained by

$$X_m^*(s,t) = \widehat{\Sigma}_m^{-\frac{1}{2}} X_m(s,t), \tag{5}$$

where $m$ is the index for a given model simulation (e.g. BCC, CCSM, ..., MPI) and $\widehat{\Sigma}_m$ is an estimated spatial covariance
matrix of $X_m(s,t)$ with $t = 850, ..., 1849$ using an exponential covariance function.

There are 283 sampling locations out of 1732 grid points. Let $f_j$ be the $j$-th PC of the sampling network from year 850 to
1849. We construct a linear model of the climate at all 1,732 locations and on $f_j$ with the decorrelated spatial fields $X_m^*(s,t)$:

$$X_m^*(s,t) = \sum_{j=1}^{K} \beta_j(s) f_j + \epsilon(s,t), \tag{6}$$

where $\epsilon(s,t)$ are white noise because $X_m^*(s,t)$ has been decorrelated. So for each fixed location $s$, we have 1,000 observations
($t = 850 - 1849$) and 1,732 different regressions will be modeled on the whole domain $D$. We set the number of EOFs to be
$K = 10$ which typically preserves about 85% of variability in the sample climate field. After we obtain $\widehat{X}_m^*(s,t)$ then we derive
$\hat{X}_m(s,t) = \widehat{\Sigma}_m^{\frac{1}{2}} \widehat{X}_m^*(s,t)$. To evaluate the model fitness, we calculate the mean squared error (MSE) as follows :

$$MSE_m = \frac{1}{1000} \Sigma_{t=850}^{1849} (\hat{X}_m(s,t) - X_m(s,t))^2. \tag{7}$$

The $MSE_m$ measures how well the sampling network represents the variability in the climate model simulation. The basic
idea is to measure how much climate variability can be recovered based on the sampled climate alone. We by no means argue
that our method is optimal for this purpose, but this MSE estimate provides a reasonable measure for the capacity of climate
sampled at the pseudoproxy locations to represent the simulated global climate in each model.

Figure 10 displays the MSE for all five climate models. The red triangles mark proxy locations and the black dots in each
plot indicate the locations with extremely high MSE ($MSE > 0.5$ and well above the third quartile as indicated by Table 3).
Because the sampling locations of the pseudoproxy network are the same across all of the models, the variation in MSE is the
result of how well the network samples the underlying covariance structures of each model simulation. Two general groups of
MSE patterns are evident in the model simulations. BCC, GISS and MPI have relatively small MSE throughout much of the

tropics and extratropics, while parts of the northern extratropics and polar region display extremely high MSE. In these model simulations, the implication is that the pseudoproxy network reasonably samples the variability in much of the global field, except for parts of the northern extratropics and polar regions. The second group comprises the CCSM and IPSL simulations. The MSE in each of these simulations is relatively high throughout the global field, with CCSM and IPSL displaying extremely high MSE in the northern extratropics and polar region and IPSL also yielding high MSE in parts of the southern extratropics and polar regions.

The sampling network in BCC well represents the temperature variability around the equator, however, it yields very high MSE in the NH extratropics. This makes the distribution of MSE associated with BCC largely skewed to the right due to the extremely large MSEs in the NH extratropics (Table 3 and Figure 10). To further aid our interpretation, we provide the maps of the first and second EOFs in Figure 11. A joint comparison between Figures 10 and 11 shows that the variability of the first and second EOFs of BCC mainly concentrates in the Northern Hemisphere where the large MSE is observed. The implication is that the pseudoproxy sampling network in BCC does not well sample variability in the NH extratropics, while the leading EOFs in BCC best represent variability over that region. This collectively further explains the poor performance of CFRs with BCC in Figure 2 (SNR=0.5).

Figure 10 also indicates that the MSE is high in CCSM and even higher for the IPSL model. Figure 11 nevertheless indicates that the main difference between CCSM and IPSL is that the CCSM simulation shows strong signal throughout the leading EOFs whereas the IPSL model only shows distinct signal in its first EOF. This helps explain the skill of CFRs associated with the IPSL PPE concentrating in its first EOF (Figure 2). On the other hand, the GISS and MPI models exhibit the smallest mean MSE, thus supporting the outstanding skill of their CFRs in reconstructing the mean. However, the performance of CFR methods, especially CCA and RIDGE, seems also additionally vulnerable to the skewness of the MSE, implied by Figure 2 (SNR=0.5). If we compare the CFR performance associated with the GISS and MPI PPEs, both TTLS and TTLH perform well but CCA and RIDGE perform better in the MPI PPE due to the relatively high skewness of MSE in the GISS model.

In summary, both the skewness of the MSE and the high MSE distribution with weak signal on the leading EOF structure together affect the skill of CFRs in all climate models. This is because large differences between the global climate and what can be sampled from the proxy network likely weakens the skill of CFRs in retaining the major mean structure of the targeted climate. In contrast, however, even if the mean MSE is high due to high variability of the temperature field, the mean structure may be well reconstructed by the CFRs if the leading EOF shows distinct signal. We also note that this analysis breaks traditional arguments about the number of degrees of freedom in the global temperature field (Hansen and Lebedeff, 1987; Briffa and Jones, 1993; Mann and Park, 1994; Jones et al., 1997; Pollack and Smerdon, 2004). These arguments are based on correlation decay distance (often defined as the e-folding distance between one grid cell and all other grid cells, e.g. Jones et al., 1997), which is assumed to be isotropic. Given an estimate of the decay distance, the associated degrees of freedom in a global field can thus be estimated. These arguments have been used to estimate the number of sampling locations that are necessary in the global field to estimate the mean climate over different timescales. Our MSE analysis nevertheless clearly indicates that covariability in global temperature fields is not isotropic and that heterogeneous sampling networks will differently sample the field variance.

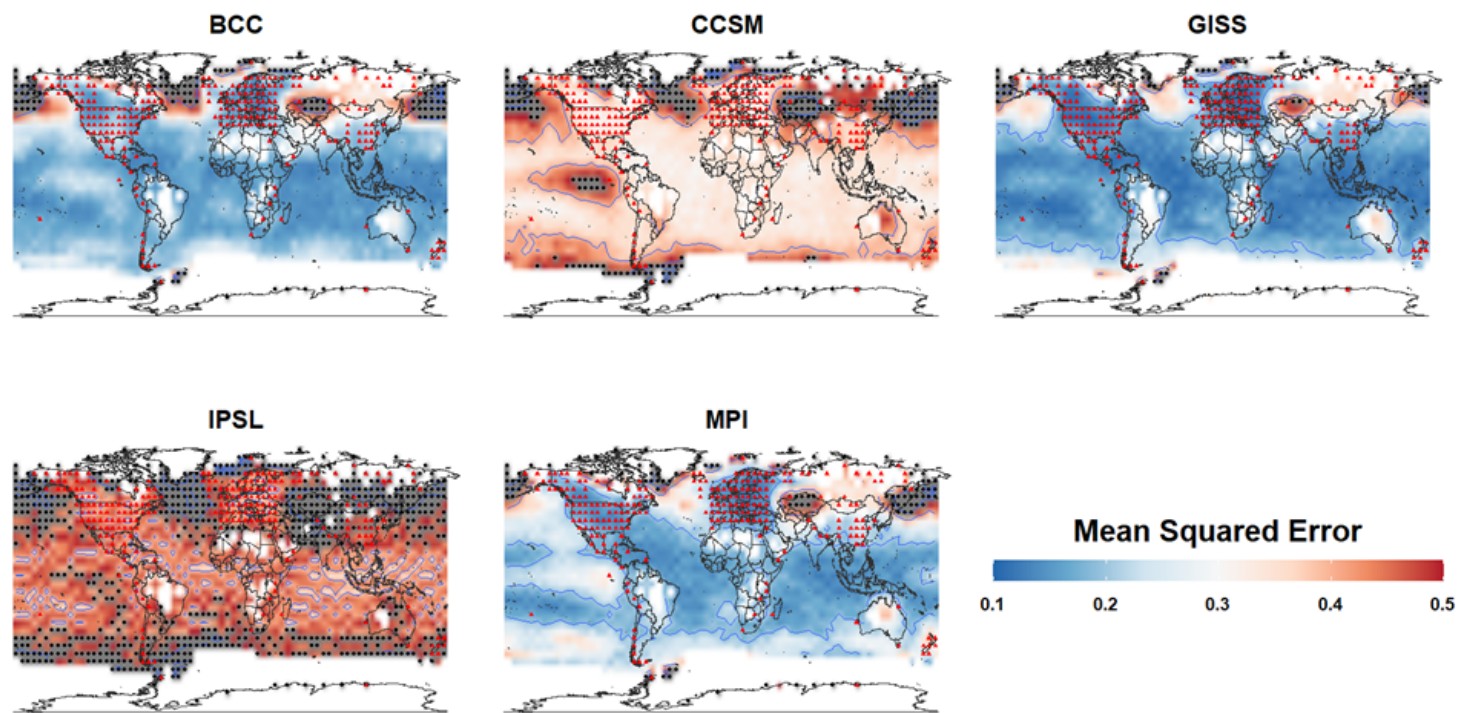

**Figure 10. Mean-square error (MSE) of sampling location regression:** The MSE of the estimated temperatures using sampling location regression is presented. The red triangles present the proxy location and the black dots indicate the extremely high MSE ($MSE > 0.5$)

**Table 3.** MSE distribution of five Climate Models

|      | MIN   | 1ST Q. | MEDIAN | MEAN  | 3RD Q. | MAX   | Skewness |
|------|-------|--------|--------|-------|--------|-------|----------|
| BCC  | 0.122 | 0.161  | 0.189  | 0.263 | 0.239  | 4.698 | 6.579    |
| CCSM | 0.290 | 0.339  | 0.370  | 0.425 | 0.430  | 2.303 | 4.162    |
| GISS | 0.101 | 0.141  | 0.171  | 0.216 | 0.229  | 2.499 | 4.965    |
| IPSL | 0.342 | 0.430  | 0.472  | 0.512 | 0.538  | 2.054 | 3.718    |
| MPI  | 0.112 | 0.169  | 0.208  | 0.250 | 0.265  | 1.788 | 3.802    |

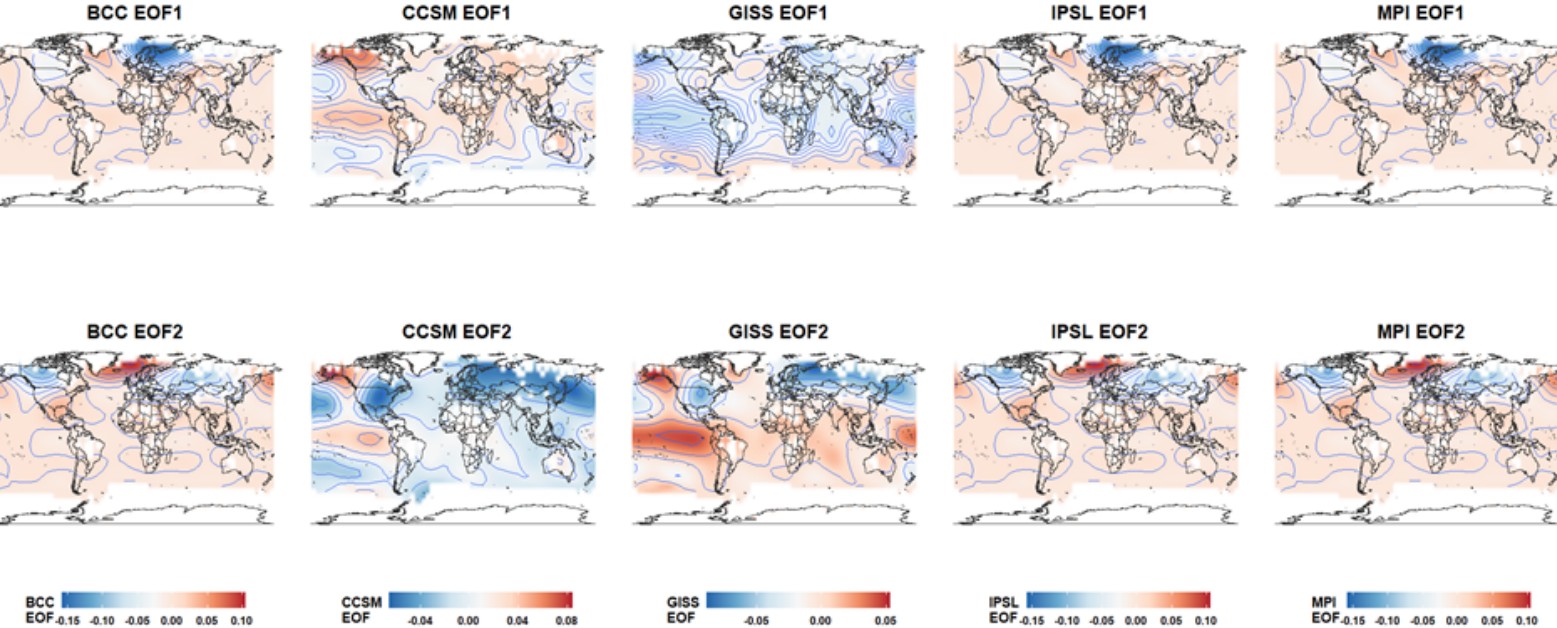

**Figure 11. 1st and 2nd EOF of climate models :** The upper and bottom panels show 1st and 2nd EOF of climate models, respectively.

## 5 Discussion and Conclusions

We have provided a comprehensive assessment of four widely-applied CFR methods in terms of their skill recovering the mean surface and covariance patterns in the targeted temperature field. Testing the mean and covariance surface jointly, based on multivariate Gaussian assumption, is a fundamental way to test the equity of the two spatiotemporal random fields as in Li and Smerdon (2012). Our motivation in this paper, however, was to separate the mean and covariance surface comparison tests for two different objectives. Comparing the mean structure evaluates how well the CFR reconstructs the annual mean temperatures and testing the equity of the covariance at ENSO dominant region evaluates how well the CFR recovers the teleconnection patterns in the fields. The assessment is conducted in the context of PPEs based on five climate model simulations spanning the 850-1995 CE interval. We have first applied the evaluation metrics presented in Li et al. (2016) and Zhang and Shao (2015) to assess the skill of each CFR with respect to the differently modeled climates. We then focused on interpreting and understanding the variability in the skill. We find that although part of the skill variability arises from the reconstruction method itself, a large part of the discrepancy in the skill across different methods is attributable to different characteristics of simulated temperature fields associated with different climate models. Our discoveries provide useful insights into the assessment and improvement of CFR methods, while the focus on the underlying characteristics of the targeted climate field make our findings relevant beyond the four methods that we have tested.

The underlying features of a targeted temperature field that can affect the performance of CFRs, as represented across the climate model simulations that we have investigated, include: (i) the characteristics of the eigenvalue spectrum, namely the amount of variance captured in the leading EOFs; (ii) the temporal stability of the leading EOFs; (iii) the representation of the climate over the sampling network with respect to the global climate; and (iv) the strength of spatial covariance, i.e. the dominance of teleconnections, in the targeted temperature field.

Our results show that the CFRs derived within the CCSM, GISS, and MPI PPEs are skillful at recovering the mean structure, whereas the CFRs associated with the BCC and IPSL PPEs exhibit large biases that are consistent with those presented in Smerdon et al. (2016). These results are likely due in part to the fact that the EOFs of the CCSM and MPI models are stable across the calibration and reconstruction periods. Additionally, the sampling network well represents the global temperature of GISS and MPI whereas is inadequate for the BCC model. This plays a key role in weakening the ability of CCA and RIDGE to reconstruct the mean of the BCC model. In terms of skill recovering the spatial covariance associated with teleconnections, the TTLS and TTLH methods outperform CCA and RIDGE, and in general CFRs derived in the CCSM PPEs outperform the CFRs associated with PPEs based on the other climate model simulations. Moreover, the CFRs of BCC and MPI show no skill in recovering the large-scale teleconnection patterns when the $SNR$ is low. For the BCC model, this low skill is also corroborated by the observation that CFRs in the BCC PPE fail to represent the variability of teleconnections in the leading EOFs of the target model in the ENSO dominant region. Within the MPI PPEs, similar challenges reconstructing the spatial covariance are likely because the teleconnection in the model simulation is already weak, as indicted by the model's low correlation between the leading five EOFs of the Niño3 region and those of the ENSO dominant region.

An important finding is that the skill of CFRs is highly associated to how well the leading EOFs in CFRs represent the targeted climate field concerning both the variability and the subspace. We find that the spectra of eigenvalues in the CCSM, GISS, and MPI models align well with their own CFRs. Among the four CFRs, the TTLS and TTLH methods better recover the eigenvalue spectrum of the targeted climate by having a large amount of variability carried by leading EOFs. In particular, CCSM exhibits the highest variability on its first few leading EOFs and this pattern is well reproduced by the corresponding EOFs in the CFRs derived from the TTLS and TTLH methods. Critically, these characteristics could be assessed for real-world data sets or through comparisons between CFRs and the observational data during the calibration and validation intervals. Such assessments are therefore strongly encouraged as additional means of testing both the likelihood of skillful reconstructions, as well as adding to a source of calibration and validation interval skill metrics.

Overall, the skill assessment we have performed using PPEs based on five climate models allows a deeper understanding of both the reconstruction methods and the characteristics of the synthetic climate fields. As we have shown, CFR assessments can vary based on the underlying spatiotemporal characteristics of the modeled target field. The ultimate goal is to evaluate and improve real-world CFRs. Based on the results of this study, the reconstruction performance can depend on the eigenvalue spectrum, the temporal stability of covariance patterns across the reconstruction and calibration intervals, the ability of sampling locations to represent the global climate characteristics, and the strength of the dominant teleconnections in the targeted climate field. A careful investigation of the characteristics of the real-world climate will help identify the likely impact of these features in CFRs derived from real proxies, as well as choose optimal reconstruction methods and proxy networks given the identified

characteristics of targeted climate fields. Although the characteristics of the real climate of course cannot be modified, our
findings will also help to define absolute limits on the skill of CFRs and thus improve their interpretations.

## Appendix

*Self normalization test*

A sampling strategy known as self normalization in the context of functional time series was developed in Zhang and Shao (2015). The self normalization is an alternative to the well-known bootstrap approach. Compared to the latter, self normalization can preserve temporal correlation without any tuning parameters. Furthermore, when used to compare two time series, including two functional time sequences, the self normalization does not require independence of the two spatiotemporal random fields. The details of the self normalization test is explained below.

Suppose $\{X_t(s)\}_{t=1}^{N_1}$ and $\{Y_t(s)\}_{t=1}^{N_2}$ are two temporally dependent functional time series. These functional time series can also be considered to be spatiotemporal random fields. Let $\mathbb{H}$ be the Hilbert space of square integrable functions over $D \subseteq \mathbb{R}^2$. For any functions $f, g \in \mathbb{H}$, the inner product between $f$ and $g$ is defined as $<f,g> = \int_D f(s)g(s)ds$, and $||f|| = <f,f>^{1/2}$ denotes the inner product of the induced norm. Define the operator $f \otimes g(\cdot) = <f, \cdot> g$ for $f, g \in \mathbb{H}$ such that for a function $h$, the operator $f \otimes g(h) = <f, h> g$ maps $h$ to $<f, h> g$. Let $L_{\mathbb{H}}^p$ be the space of $\mathbb{H}$-valued random variables $X$ such that $E||X||^p < \infty$ for some $p > 0$.

Assuming that the spatiotemporal random fields are second-order stationary in time, we define $\mu_X(s) = E\{X_t(s)\}$ and $\mu_Y(s) = E\{Y_t(s)\}$ as their mean functions over $s \in D$. We consider testing the following hypothesis,

$$H_0 : \mu_X = \mu_Y \quad \text{vs.} \quad H_a : \mu_X \neq \mu_Y, \tag{A.1}$$

where $\mu_X \neq \mu_Y$ is equivalent to $||\mu_X - \mu_Y|| > 0$.

Below we will define the recursive sample mean function which preserves the temporally dependent structure:

$$\mu_{X,m} = \frac{1}{m} \sum_{t=1}^{m} X_t \quad \text{and} \quad \mu_{Y,n} = \frac{1}{n} \sum_{t=1}^{n} Y_t,$$

where $1 \leq m \leq N_1$ and $1 \leq n \leq N_2$. The pooled sample covariance operator is defined as

$$\widehat{C}_{XY} = \frac{1}{N} \Big[ \sum_{t=1}^{N_1} \{X_t - \hat{\mu}_{X,N_1}\} \otimes \{X_t - \hat{\mu}_{X,N_1}\} + \sum_{t=1}^{N_2} \{Y_t - \hat{\mu}_{Y,N_2}\} \otimes \{Y_t - \hat{\mu}_{Y,N_2}\} \Big],$$

where $N = N_1 + N_2$ is the total time length for two random fields. The eigenvalues and eigenfunctions corresponding to $\widehat{C}_{XY}$ are denoted by $\{\hat{\lambda}_{XY}^j\}$ and $\{\hat{\phi}_{XY}^j\}$. Then we define a sequence of vectors consisting of the projected (recursive) mean differences on the first $K$ eigenfunctions:

$$\hat{\psi}_k = \Big( <\hat{\mu}_{X,\lfloor kN_1/N \rfloor} - \hat{\mu}_{Y,\lfloor kN_2/N \rfloor}, \hat{\phi}_{XY}^1 >, ..., <\hat{\mu}_{X,\lfloor kN_1/N \rfloor} - \hat{\mu}_{Y,\lfloor kN_2/N \rfloor}, \hat{\phi}_{XY}^K > \Big)^T$$

for $2 \leq k \leq N$, where $\lfloor W \rfloor$ is the largest integer not greater than $W \in \mathbb{R}$. Then the test statistic for hypothesis (A.1) is

$$TS(K) = N\hat{\psi}_N^T V_\psi^{-1}(K)\hat{\psi}_N$$

where $V_\psi(K) = \frac{1}{N^2}\sum_{k=1}^{N}k^2(\hat{\psi}_k - \hat{\psi}_N)(\hat{\psi}_k - \hat{\psi}_N)^T$. The parameter $K$ is a user-chosen number that determines the number of eigenfunctions to be used in the test and is associated with the percentage of total variation with respect to the pooled sample covariance.

The $K$-length vector $\hat{\psi}_k$ consists of projected differences, with the $j$th element being projected onto the $j$th eigenfunction $\hat{\phi}_{XY}^j$. The index $k$ for $\hat{\psi}_k$ indicates the $k$th paired difference between the recursive estimates of mean functions. Because of these recursive estimates that are the kernel of the self-normalization technique, we allow individual data to be temporally correlated and moreover the two data sets to be correlated if additionally assuming $N_1/N_2 \to 1$.

The pivotal limiting distribution of $TS(K)$ is derived in Zhang and Shao (2015). Here we summarize how to calculate the p-value. Define $B_q(r)$ as a $q$-dimensional vector of independent standard Brownian motions. Let $W_q = B_q(1)^T J_q^{-1} B_q(1)$, where $J_q = \int_0^1 \{B_q(r) - rB_q(1)\}\{B_q(r) - rB_q(1)\}^T dr$, then $TS(K)$ converges to $W_K$. The empirical distribution of $W_q$ for any $q$ can be obtained numerically by approximating the standard Brownian motion with the standardized partial sum of i.i.d standard normal random variables.

### Availability of data and codes

Codes and data to reprodue the skill assessment comparison test are available at GitHub (https://github.com/syun0925/CFR.git).

### Acknowledgements

The authors thank the editor and the referees for constructive suggestions that have improved the content and presentation of this article. This work was supported by the National Science Foundation grant AGS-1602845. Li was also partially supported by the NSF-DMS-1830312 and NSF-DGE-1922758.

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
