# Peer review of "A pseudoproxy assessment of why climate field reconstruction methods perform the way they do in time and space"

_Climate of the Past, 2020_

## Referee Comment (RC1)

**Review Report on "A pseudoproxy assessment of**

**why climate field reconstruction methods perform the way they do in time and space"**

The manuscript assesses four different climate field reconstruction (CFR) methods, two of which are based on spectral domain analysis, and the other two are based on ridge regression and canonical correlation analysis, respectively. These methods are evaluated through pseudoproxy experiments (PPE) with five different GCMs (BCC, CCSM, GISS, IPSL, and MPI), using singular value decomposition-based hypothesis tests on the mean and covariance functions. The hypothesis test results and the follow-up analyses show that the reconstruction performances are affected by (i) how well the overall patterns in a GCM simulation are captured by the leading EOFs, (ii) how temporally stable the leading EOFs are, and (iii) how the sampled locations are representative with respect to the global climate patterns.

I think the analysis presented in the manuscript is highly through and carefully done, and it sheds lights on the factors that can negatively affect the performance of the existing CFR approaches and how CFR methods can be improved in the future. The suggestions below are mostly on improving the presentation and fixing some typos/incorrect statements:

1. In lines 215- : I think the fact that $X$ is the synthetic truth and $Y$ is the CRF result should be introduced earlier so that readers can understand Sections 3.1.1 and 3.1.2 more easily. In the current format, readers should wait until the end of 3.1.2 to find this out, while wondering how $X$ and $Y$ are chosen.

2. Line 226 "the correlation of climate observations …": I would say "the correlation and the variance of climate observations …"

3. Line 232: There are redundant parentheses.

4. Lines 235- : I think it is worth mentioning possible computational issues in conducting singular value decomposition when the data size is large, perhaps mentioning the computational complexity is in $O(\min(mn^2, m^2n))$.

5. Line 244: I am wondering if $k^2$ in the equation for $V(d)$ is a typo. If not, please add a sentence that explains why $k^2$ is needed.

6. Lines 255-257: Perhaps what the authors meant were "In other words, a p-value close to 0 indicates the difference between modeled and reconstructed fields is statistically significant against this null hypothesis, while p-values close to 1 indicates the difference could be explained by random chance."

7. Lines 258-260: I think the authors need more motivation on why they are focusing on the leading five principal components here. My guess is that the leading principal components are mostly large-scale features, which are usually the main interest when studying the changes in spatial patterns (e.g. ENSO). I think authors want to clarify this point here.

---

## Author Comment (AC1)

In the following responses, we reproduce the reviewers' comments in italics and include our detailed responses in bold text thereafter.

**Response to Reviewer 1**

*The manuscript assesses four different climate field reconstruction (CFR) methods, two of which are based on spectral domain analysis, and the other two are based on ridge regression and canonical correlation analysis, respectively. These methods are evaluated through pseudoproxy experiments (PPE) with five different GCMs (BCC, CCSM, GISS, IPSL, and MPI), using singular value decomposition based hypothesis tests on the mean and covariance functions. The hypothesis test results and the follow up analyses show that the reconstruction performances are affected by (i) how well the overall patterns in a GCM simulation are captured by the leading EOFs, (ii) how temporally stable the leading EOFs are, and (iii) how the sampled locations are representative with respect to the global climate patterns. I think the analysis presented in the manuscript is highly thorough and carefully done, and it sheds light on the factors that can negatively affect the performance of the existing CFR approaches and how CFR methods can be improved in the future. The suggestions below are mostly on improving the presentation and fixing some typos/incorrect statements:*

1. *In lines 215- : I think the fact that $X$ is the synthetic truth and $Y$ is the CFR result should be introduced earlier so that readers can understand Sections 3.1.1 and 3.1.2 more easily. In the current format, readers should wait until the end of 3.1.2 to find this out while wondering how $X$ and $Y$ are chosen.*

   **We appreciate this comment and will modify the text to reflect this fact.**

2. *Line 226 "the correlation of climate observations": I would say "the correlation and the variance of climate observations".*

   **We will modify the text in accordance to this comment in the revised manuscript.**

3. *Line 232: There are redundant parentheses.*

   **We will remove the redundant parentheses.**

4. *Lines 235: I think it is worth mentioning possible computational issues in conducting singular value decomposition when the data size is large, perhaps mentioning the computational complexity is in $O(min(mn^2, m^2n))$.*

   **We agree that mentioning the computational time and complexity of the singular value decomposition is a good idea. This will be added to the revised manuscript.**

5. *Line 244: I am wondering if $k^2$ in the equation for $V(d)$ is a typo. If not, please add a sentence that explains why $k^2$ is needed.*

   **The $k^2$ in the equation for $V_\alpha(d)$ is not a typo. Because the recursive $\hat{\alpha}_k$ is estimated based on the recursive sample comprising observations from 1 to $\lfloor k/2 \rfloor$, the weight $k^2$ in $V_\alpha(d)$ is to account for the sample size used to estimate $\alpha_k$. This procedure is capable of incorporating the temporal dependence. Details will be added in the revised manuscript.**

6. *Lines 255-257: Perhaps what the authors meant were "In other words, a p-value close to 0 indicates the difference between modeled and reconstructed fields is statistically significant against this null hypothesis, while p-values close to 1 indicates the difference could be explained by random chance."*

   **The reviewer is correct and we appreciate that this mistake was pointed out. We will fix this statement in the revised manuscript.**

7. *Lines 258-260: I think the authors need more motivation on why they are focusing on the leading five principal components here. My guess is that the leading principal components are mostly large-scale features, which are usually the main interest when studying the changes in spatial patterns (e.g., ENSO). I think the authors want to clarify this point here.*

   **The reviewer's conjecture is correct. The leading 5 EOFs consist of more than 80% of the total variability and largely represent the dominant spatial patterns of the random fields. We therefore compare the features of the spatiotemporal fields on these leading five principal components. We will clarify this point in the revised manuscript.**

---

## Author Comment (AC2)

In the following responses, we reproduce the reviewers' comments in italics and include our detailed responses in bold text thereafter.

**Response to Reviewer 2**

1. *The biggest issue I have with the paper is trying to figure what was done in the methods. Obviously, we need this kind of comparison in climatology and it is extremely important. I didn't have any issues with the scientific prose in the paper, but I would like the authors to be much more pedagogical in their methodological exposition so that folks can discern what has been done. I did not get anything out of the brief description of the four methods in Section 2. Hence, I was hoping that Section 3 would alleviate these concerns; alas, it did not.*

   **The details of the four different CFR methods employed in our manuscript are widely reported across many different publications, which we summarize in Section 2. We nevertheless will be happy to further explain the methods that underlie the employed reconstructions to better situate the reader**

2. *In Section 3, some things are not stated that need to be: are the $X$ and $Y$ processes independent (I think so)? It seems you are assuming a constant mean in time $t$ (which is not likely true) and that the covariance function of the spatial fields at each time have the same structure (I can buy this). I also want to know if you are assuming that the fields are Gaussian.*

   **We will further elucidate the following points in the manuscript.**

   - **Our method does not require $X$ and $Y$ to be independent. This is one of the advantages of our functional data analysis methods.**
   - **We do not assume a constant mean in $t$. We removed a common trend that is a constant at $t$, and then we allow the mean of the detrended data to be spatially varying. We indeed assume the spatial covariance function follows the same structure at each time.**
   - **Our method does not require the random fields to be Gaussian, though the climate model data can be approximated by a Gaussian random field. In the case of Gaussian random fields, our test for mean and**

covariance is equivalent to the test for distribution because the mean and covariance determine the Gaussian distribution, but this equivalence cannot be generalized to other distributions.

3. *In testing for whether the means of the two processes are the same, why would we not just look at the average (overall spatial locations and times) and use asymptotic normality to test whether these $X$ minus $Y$ averages have a zero mean? This just works with differences $\Delta(t, s) = X(t, s) - Y(t, s)$. Then you don't have to assume the mean is constant....it subtracts to zero under the null. You can easily estimate the variances of the average $\Delta$ value assuming a null that the two fields have the same covariance structure. This seems to be the fundamental way to handle the two sample equality issue in general abstract spaces. I'm guessing that what you've done can be justified, but it would seem that I have to go to your past papers to dig this up. I just have this uneasy feeling that the EOF approach is needlessly complicated.*

We would like to clarify that we did not assume constant mean for the whole spatial domain. Instead, we allow the mean to be spatially varying and we aim to test whether $\mu_X(s) = \mu_Y(s)$. We removed the common trend from $X$ and $Y$, which is calculated as the global average at each year based on both data sets. This detrending procedure will not affect our spatial mean test.

Regarding the reviewer's suggestion, we agree it is the most natural approach. We indeed carried out a very similar idea in Li and Smerdon (2012) by centering, scaling and decorrelating $X$ and $Y$ using their common mean and common covariance matrix, and then evaluating whether the two post-processed data sets followed the same distribution. However, working on that project made us realize several drawbacks of this seemingly rigorous method, and motivated us to seek a more robust and flexible method resulting in our follow-up work Li et al. (2016). The drawbacks can be summarized as: 1) temporal dependence is not taken into account; 2) $X$ and $Y$ are assumed to be independent; and 3) the result is sensitive to misspecification of the spatial covariance function. We later turned to the functional data method in Li et al. (2016) which was used in our current manuscript for the following reasons: 1) it allows dependence between $X$

and $Y$ and temporal correlation within each data; 2) it is nonparametric, thus there is no concern for model misspecification; and 3) it enables an assessment of the discrepancies between $X$ and $Y$ at different directions or subspaces.

4. *I also can't rationalize why I need to use the data from times 1 to k in various places. Seems I should use all N times once.*

We apologize for the confusion. We used the data from times 1 to $k$ because we employed a self-normalization approach to generate samples for the variance estimation. Self normalization is analogous to bootstrap or subsampling, but bootstrap tends to break the temporal correlation and subsampling needs to choose the optimal subsample size and the subsample form. Self normalization for temporally correlated data was developed by Zhang and Shao (2015) for generating recursive samples for the parameter estimates in which the variance is the variable of interest. The recursive samples are obtained by drawing samples from time 1 to $k$, $k = 2, ...N$, meaning each time the new sample is formed by expanding the previous sample by adding the current observation. Self normalization is also tuning-parameter free and allows for temporal correlation. More details can be found in Zhang and Shao (2015).

5. *It would seem to me that we want to test whether the means are the same and the covariances are the same in tandem. Not either the mean is the same or the covariance is the same separately, but to test both in tandem. So why not set $\Delta(t, s) = X(t, s) - Y(t, s)$ and work with these differences as above. If the means are the same, the mean of the $\Delta$ process is identically zero at all times and spatial locations. Then we could stack the $\Delta(t, s)$ in a giant vector — call it $V$ — over all spatial locations and time points. Now, if we could get the covariance matrix of all components in this giant vector — call it $\Sigma$ — we would just look at $\Sigma^{-1/2}V$. This quantity would be composed of IID N(0,1) variates if the original fields are Gaussian. And it is easy to test whether data is $IID \ N(0, 1)$ by a plethora of methods (QQ plots, Kolmogorov-Smirnov, chi-squared tests, etc). To estimate $\Sigma$, grab your favorite space-time covariance estimators to estimate both the $X$ covariance and the $Y$ covariance structures in time and space. Call these estimates*

$\Sigma_X$ *and* $\Sigma_Y$, *respectively. Let* $\Sigma^* = (\Sigma_X + \Sigma_Y)/2$ *be the common estimate under the null that the two processes have the same covariances and are independent. Now just use* $Cov(\Delta(t,s), \Delta(t',s')) = 2$ *times the corresponding entry in the matrix* $\Sigma^*$*. Then I think it's game over: you've tested both hypotheses at the same time.*

**We once again fully agree with the reviewer that their suggestion is most natural. As mentioned in our response for bullet point 3, we have tried a similar idea in Li and Smerdon (2012). The method works to some degree, but it is unsatisfactory in several ways as we listed in point 3. The temporal correlation in climate data, the possible dependence between the synthetic ($X$) and CFR ($Y$) data, and the complex correlation structure of the global climate data that can challenge the validity of any stationary and parametric covariance function, gave us impetus to seek more flexible and robust methods. Hence, we have employed the functional data method in our manuscript. Another benefit for principal component based methods is that noise will be filtered out in the analysis. This is very important for our hypothesis testing, because we are analyzing very high dimensional data for which noise can dominate the result and lead to misleading conclusions.**

**References**

Li, B. and J. E. Smerdon (2012). Defining spatial comparison metrics for evaluation of paleoclimatic field reconstructions of the common era. *Environmetrics 23*(5), 394–406.

Li, B., X. Zhang, and J. E. Smerdon (2016). Comparison between spatio-temporal random processes and application to climate model data. *Environmetrics 27*(5), 267–279.

Zhang, X. and X. Shao (2015). Two sample inference for the second-order property of temporally dependent functional data. *Bernoulli 21*(2), 909–929.

---

## Referee Report (RR1)

Referee report on a revision of *A pseudoproxy assessment of why climate field reconstruction methods perform the way they do in time and space*, by Yun, Smerdon, Li, and Zhang.

**Overall Comments**: As much as I like the topic of this paper, I don't find the revision to be much more informative than the original. The typos are now gone, but new issues arise and the authors still delegate many important matters to citations. I am a statistician, trying to learn how to conduct these tests, and this paper is incredibly vague. I could not reproduce any of the results from what is given in the paper. I do not want to see another version of this. Maybe some of my troubles lie with the Copernicus' journal's editorial process, but it is frustrating to see virtually the same paper again.

**Specific Comments**:

1. I am still unclear how the hypotheses are tested. I believe that the authors are getting $p$-values from some sort of self normalization procedure, but this is never made clear. I previously thought that they were coming from the chi-squared distribution, but now I doubt this (lines 259 and circa 275). But if true, doesn't self normalization need to be discussed? Since I am unfamiliar with this technique, what hope do climate scientists have? This is my frustration with the paper: I do not know why we are doing what is being done.

2. At the centre of the tests, I still don't understand why we are projecting onto eigenfunctions. If I want to test whether the mean is the same from two samples at a fixed site, I look at the difference between the univariate averages — univariate asymptotic normality comes up. If we want to examine all sites in $\mathcal{D}$ simultaneously, a vector of mean differences arise and multivariate asymptotic normality arises. The authors provide some words on this, but hardly anything swaying.

3. There still isn't anything that I see in the paper that tests for both equality of means and autocovariances simultaneously.

4. Section 2.2 seems new, but its notation is bad! First, you are denoting variants of quantities with a prime, and mixing this in equations where $T$ denotes transpose. Compounding this, you have a variable named $T$! Matrices and vectors are not bolded. There are quantities like $P^r$ related to $P'$ (why suppress the prime?). It took me an hour to deconvolve this simple section!

5. Grammatically, the paper is pretty good. Nonetheless, there are a few spots where articles are abused or there is awkwardness. For one such example, the first line in the abstract should probably start with "This paper derives". And spatiotemporal really still needs a dash to be Oxford compliant.

6. I apologize for being so picky, but your paper seems like a black-box for climate scientists to follow rather than something informative.

---

## Author Response (AR2)

We want to thank the editor for his guidance in addressing each specific comment. In the following responses, we reproduce the reviewers' comments in italics and include our detailed responses in bold text thereafter.

**Overall Comments:** *As much as I like the topic of this paper, I don't find the revision to be much more informative than the original. The typos are now gone, but new issues arise and the authors still delegate many important matters to citations. I am a statistician, trying to learn how to conduct these tests, and this paper is incredibly vague. I could not reproduce any of the results from what is given in the paper. I do not want to see another version of this. Maybe some of my troubles lie with the Copernicus' journal's editorial process, but it is frustrating to see virtually the same paper again.*

**We address the specific comments of the reviewer in the sections below. We nevertheless wish to push back on the contention that our last revision was "was virtually the same paper," relative to what we originally submitted. To the contrary, we submitted a comprehensive response to the reviewers and a heavily edited manuscript. We therefore respectfully disagree with the reviewer's assessment, which is easily refuted by a quick scan of the track-changes version of the previous submission.**

**Specific Comments:**

1. *I am still unclear how the hypotheses are tested. I believe that the authors are getting p-values from some sort of self normalization procedure, but this is never made clear. I previously thought that they were coming from the chi-squared distribution, but now I doubt this (lines 259 and circa 275). But if true, doesn't self normalization need to be discussed? Since I am unfamiliar with this technique, what hope do climate scientists have? This is my frustration with the paper: I do not know why we are doing what is being done.*

   **Because the self normalization test is a well developed technique, we did not include many details in the paper to avoid redundancy. For the convenience of readers, however, we now provide the details of the self-normalization procedure in the new Appendix of the revised manuscript.**

*2. At the centre of the tests, I still don't understand why we are projecting onto eigen-functions. If I want to test whether the mean is the same from two samples at a fixed site, I look at the difference between the univariate averages – univariate asymptotic normality comes up. If we want to examine all sites in $\mathcal{D}$ simultaneously, a vector of mean differences arise and multivariate asymptotic normality arises. The authors provide some words on this, but hardly anything swaying.*

As we noted in our previous response, if we just want to test whether the mean is the same from two samples at a fixed site, the two sample $t$ test (or the asymptotical normality test) would be sufficient. To examine the difference at all sites in $\mathcal{D}$ simultaneously, however, the chi-square test based on the multivariate normality of observations at all sites has several practical and technical issues. The chi-square test treats the mean at each site equally such that noise can dominate the testing results, whereas our current test aims to examine the difference at major modes of the climate fields. The projection onto EOFs in our method helps filter out the noise and thus allows us to focus on the subspace containing the leading dynamical climate pattern. Additionally, our method can test both the mean and covariance, while chi-square only tests the mean. One technical challenge for the multivariate normality test for high dimensional data is to estimate the variance and covariance function of the multivariate process over the globe. Climate data measure a highly heterogeneous process and any parametric models will be limited in their ability to capture the dependency structure of a global process. For example, our previous paper (Li and Smerdon, 2012) demonstrated that the chi-square test is sensitive to the misspecification of the covariance function. Our current method is purely non-parametric and thus free of the risk of model misspecification. Another technical issue for the multivariate normality test is that the method requires the two multivariate data to be independent, a condition that obviously does not hold for our data. In contrast, the method that we have applied allows for dependence between the two data sets. All of these points relating to the comparison between our method and the more traditional multivariate normality (chi-square test) method are now even more plainly addressed in

**Lines 82-87 of the revised manuscript.**

3. *There still isn't anything that I see in the paper that tests for both equality of means and autocovariance simultaneously.*

   We agreed that tests for both equality of means and autocovariance simultaneously are a natural way to evaluate the discrepancies between two climate fields, and that is exactly what Li and Smerdon (2012) and Li et al. (2016) did in their papers using either a parametric method or a functional data method. Because our current method aligns with Li et al. (2016), we can easily implement their joint test on our data, but obtaining a single p-value is not the focus of this paper. Instead, we are interested in dissecting each segment of the testing results to understand the mechanism behind the differences across the CFRs. A joint test for both the equality of means and autocovariance therefore does not add value to these interests. To explicitly explain why we did not perform the joint test, we have added the below text to the revised manuscript at Lines 263-266:

   *"Another available test for evaluating the difference between two climate fields is to combine hypotheses (i) and (ii) into one single test, as in Li and Smerdon (2012) and Li et al. (2016). We omit this joint test because the focus of this paper is to understand why the mean and covariance in a reconstructed field behave differently. Thus, each individual test is sufficient and more pertinent for such a purpose."*

   This explanation is also reiterated at Lines 596-601 in the Discussion and Conclusion Section.

4. *Section 2.2 seems new, but its notation is bad! First, you are denoting variants of quantities with a prime, and mixing this in equations where $T$ denotes transpose. Compounding this, you have a variable named $T$! Matrices and vectors are not bolded. There are quantities like $P^r$ related to $P'$ (why suppress the prime?). It took me an hour to deconvolve this simple section!*

   We have revised this section with the following changes:

- Standardized matrices of $P$ and $T$ are now denoted as $P_{std}$ and $T_{std}$, respectively.

- Reduced rank representations of $P_{std}$ and $T_{std}$ are now denoted as $P_{std}^r$ and $T_{std}^r$, respectively.

We use a superscript $T$ to note the transpose of a matrix, which is a common notation in the statistical literature and is clearly distinguishable from where the temperature matrix $T$ is used.

In our experience, the guidelines for matrix notation vary with different journals. Some journals require bold text for matrix variables, while some specifically require that they are not bold. Regardless of the journal, however, all notation should be consistent throughout a given paper. The notation of matrices is consistently not bold throughout our manuscript and for now we maintain this convention. If 'Climate of the Past' suggests that we use bold matrix notation throughout our manuscript, we will be happy to do so.

5. *Grammatically, the paper is pretty good. Nonetheless, there are a few spots where articles are abused or there is awkwardness. For one such example, the first line in the abstract should probably start with "This paper derives". And spatiotemporal really still needs a dash to be Oxford compliant.*

We leave this to the proofreading and copy editing, as suggested by the editor.

6. *I apologize for being so picky, but your paper seems like a black-box for climate scientists to follow rather than something informative.*

This characterization is off base. Our paper is comprehensively cited, allowing any reader to discover the specifics of the method that was applied. The purpose of our paper, however, is to use existing methods to elucidate why and how CFR methods perform the way they do. It is therefore entirely appropriate to provide citations as the principal background on the employed methods, while complementing the discussion with methodological

summaries as we have done in our paper. While we believe this approach is sufficient and concise, we now provide even more details on the statistical methodology in a new Appendix within the revised manuscript. We also will publish access to the codes used to perform our analyses on the GitHub site now listed in the revised manuscript.

**References**

Li, B. and J. E. Smerdon (2012). Defining spatial comparison metrics for evaluation of paleoclimatic field reconstructions of the common era. *Environmetrics 23(5)*, 394–406.

Li, B., X. Zhang, and J. E. Smerdon (2016). Comparison between spatio-temporal random processes and application to climate model data. *Environmetrics, 27(5) 27*, 267–279.